# Physics-informed transformers for electronic quantum states

**João Augusto Sobral** [1] ✉, **Michael Perle** [2] **& Mathias S. Scheurer** [1]

Neural-network-based variational quantum states, particularly autoregressive models, are powerful tools for describing complex many-body wave functions. However, their performance depends on the computational basis chosen and they often lack physical interpretability. We propose a modified variational Monte-Carlo framework which leverages prior physical information to construct a complete computational many-body basis containing a reference state that serves as a rough approximation to the true ground state. A Transformer is used to parametrize and autoregressively sample corrections to this reference state, giving rise to a more interpretable and computationally efficient representation of the ground state. We demonstrate this approach in a fermionic model featuring a metal-insulator transition by employing Hartree-Fock and a strong-coupling limit to define physics-informed bases. We also show that the Transformer's hidden representation captures the natural energetic order of the different basis states. This work paves the way for more efficient and interpretable neural quantum-state representations.

Neural quantum states (NQS) have been successfully used within Variational Monte Carlo (VMC) to describe highly accurate and flexible parametrizations of the ground state wavefunction of a variety of many-body physical systems[1–7]. Parallel developments have expanded NQS capabilities to capture excited states[8,9], while improvements of the stochastic reconfiguration method[10,11] have enhanced both the scalability and accuracy of these variational ansätze. Recently, hybrid approaches which integrate NQS with experimental or computational projective measurements in a pre-training stage[12–15], or quantum-classical ansätze[16,17] have also shown substantial VMC performance improvements.

Neural autoregressive quantum states (NAQS), which are based on the idea of efficiently parameterizing joint distributions as a product of conditional probabilities, have acquired substantial attention due to their general expressiveness and ability to perform efficient and exact sampling[5,18]. Recurrent Neural Networks[19,20] and Transformers[21,22] constitute prominent examples of autoregressive architectures commonly used as variational ansätze[23–27]. Transformer quantum states (TQS), in particular, have proven effective in providing highly accurate representations of ground states in frustrated magnetism[27,28], quantum chemistry[29,30], and Rydberg atoms[31], while also holding promise

for interpretability within the context of the self-attention mechanism[32–35].

Despite their versatility, NQS effectiveness may still depend on the basis in which the Hamiltonian is represented. For instance, Robledo-Moreno et al.[36] demonstrated that variationally optimized single-particle orbital rotations can significantly improve the accuracy of calculated observables. Furthermore, NQS wave function representations may lack direct physical interpretability, e.g., with respect to the relative frequency of sampled states from the Hilbert space. This contrasts with post-Hartree-Fock (HF) methods in quantum chemistry, such as coupled cluster theory[37], where corrections are naturally interpreted as single or double excitations to the HF state.

We present a modified VMC approach that simultaneously addresses these aspects. Although the method is architecture-agnostic, we demonstrate its effectiveness using a Transformer-based[26,27] framework. As a first step, an effective theory−a simplified solvable model, $\hat{H}_0$, that aims at capturing the essential physics in specific parameter regimes of the full Hamiltonian $\hat{H}$−is introduced and its spectrum defines the computational basis (see also Fig. 1a); for concreteness, we here use two examples−the basis that diagonalizes the Hamiltonian in the mean-field approximation and a natural basis in the

[1]Institute for Theoretical Physics III, University of Stuttgart, Stuttgart, Germany. [2]Institute for Theoretical Physics, University of Innsbruck, Innsbruck, Austria. ✉e-mail: joao.sobral@itp3.uni-stuttgart.de

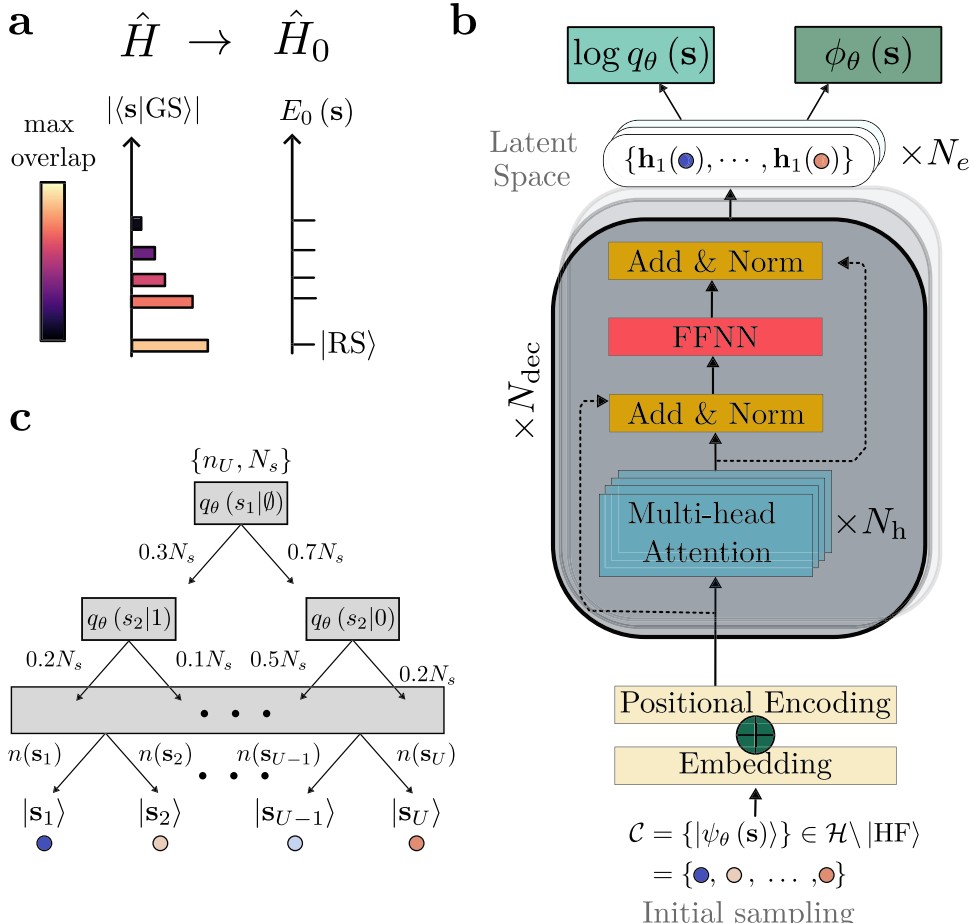

**Fig. 1 | General methodology. a** First, we choose an effective theory $\hat{H}_0$ approximating the target Hamiltonian $\hat{H}$, e.g., via a mean-field approximation or by taking the strong-coupling limit. We use the groundstate $|RS\rangle$ and excited states $|s\rangle$ of $\hat{H}_0$ to define a physics-informed, interpretable basis for the Transformer (**b**) in Equation (4); as long as the dominant weight of the ground state of $\hat{H}$ is in the low-energy part of the spectrum $E_0(s)$ of $\hat{H}_0$, this further improves sampling efficiency and the expressivity of the ansatz. **c** We sample the states $s$ using the batch-autoregressive sampler[57,58,63]. It is controlled by the batch size $N_s$ and the number of partial unique strings $n_U$, and directly produces the relative frequencies $r(s)$ associated with each state in a tree structure format. Back to (**b**), the states $s$ are then mapped to a high-dimensional representation of size $d_{\text{emb}}$ and passed through $N_{\text{dec}}$ decoder-layers[26], containing $N_h$ attention heads, which produce correspondent representations $h(s) \in \mathbb{R}^{d_{\text{emb}}}$ in latent space. In Supplementary Note B6 we explain how these parameters are chosen. As discussed in the main text, the wavefunctions $\psi_\theta(s) = \sqrt{q_\theta(s)}e^{i\phi_\theta(s)}$ can be directly obtained from these vectors. A new set of states $\mathcal{C}$ is then obtained, according to the updated $q_\theta(s)$, and the process is repeated until the convergence of $\{\theta, \alpha\}$ according to Equation (5).

limit of strong interactions of our model. Both of these bases contain a "reference state" (RS) which is a candidate for an approximate description of the ground state of the system. In the case of the mean-field approximation, the RS just corresponds to the Hartree-Fock (HF) ground state. Meanwhile, for the second basis, the RS is the exact ground state at strong coupling. We explicitly parametrize the weight of the RS using a single parameter $\alpha \in \mathbb{R}$ while the Transformer network focuses on describing the corrections to it. Apart from enhancing convergence, $\alpha$ is convenient as it directly quantifies how close the many-body state is to the interpretable RS. We emphasize that this approach (as opposed to, e.g., coupled cluster methods) is not biased toward favoring states close to the RS or, equivalently, $\alpha$ near 1. In fact, we demonstrate explicitly that the technique leads to a vanishingly small weight of the RS should the latter not be a good approximation to the true ground state. In addition, for example, in the HF basis, the remaining basis states have a natural interpretation as being associated with a certain number of particle-hole excitations in the HF bands. This produces a natural energetic hierarchy that we also recover both in their relative weight and hidden representation of the Transformer's parameterization of the many-body ground state.

To exemplify this methodology, we use a one-dimensional interacting fermionic many-body model in momentum space. This model features an exactly solvable strong-coupling limit, which is used to define the strong-coupling basis mentioned above. Moreover, it exhibits a finite regime where integrability is no longer apparent, showing clear differences between exact diagonalization (ED) and HF, where corrections to mean-field treatments become significant.

Our results demonstrate that when the true ground state is close to a product state (the strong coupling limit), the HF basis (strong coupling basis) guides the TQS to converge to a variational representation with two key characteristics: (i) the number of states required for an accurate ground state representation only involves a fraction of the total Hilbert space which is learned and efficiently sampled from by the Transformer; (ii) the states self-organize hierarchically by their statistical weights, with a clear physical structure on latent space, naturally representing excitations on top of the RS. Finally, we show how these features contrast sharply with a generic basis, which generically requires an exponentially large amount of states, hindering scalability and the identification of dominant corrections to mean-field treatments.

# Results

## General formalism

Our central goal is to determine the ground state of a general inter-acting fermionic Hamiltonian $\hat{H}$ given by

$$\hat{H} = \sum_{a,b,\boldsymbol{k}} d^\dagger_{\boldsymbol{k},a} h_{a,b}(\boldsymbol{k}) d_{\boldsymbol{k},b} + \\ + \sum_{\substack{a_1,a_2,b_1,b_2 \\ \boldsymbol{k}_1,\boldsymbol{k}_2,\boldsymbol{k}_3,\boldsymbol{k}_4}} d^\dagger_{\boldsymbol{k}_1,a_1} d^\dagger_{\boldsymbol{k}_2,a_2} d_{\boldsymbol{k}_3,b_2} d_{\boldsymbol{k}_4,b_1} V^{\boldsymbol{k}_1,\boldsymbol{k}_2,\boldsymbol{k}_3,\boldsymbol{k}_4}_{a_1,a_2,b_2,b_1}, \tag{1}$$

where $d^\dagger_{\boldsymbol{k},a}$ and $d_{\boldsymbol{k},a}$ are fermionic, second quantized creation and annihilation operators with momentum $\boldsymbol{k}$, and indices $a, b, \ldots$ indicate additional internal degrees of freedom of the system, such as spin and/or bands. The one and two-body terms are determined by $h_{a,b}(\boldsymbol{k})$ and $V^{\boldsymbol{k}_1,\boldsymbol{k}_2,\boldsymbol{k}_3,\boldsymbol{k}_4}_{a_1,a_2,b_2,b_1}$, respectively; although not a prerequisite for our method, we assume translational invariance for notational simplicity.

A first approximation to the ground state of Equation (1) can be provided by HF[38,39]; restricting ourselves to translation-invariant Slater-determinants, HF can be stated as finding the momentum-dependent unitary transformations $U_{\boldsymbol{k}}$ of the second-quantized operators,

$$\bar{d}_{\boldsymbol{k},p} = \sum_a (U_{\boldsymbol{k}})_{p,a} d_{\boldsymbol{k},a}, \tag{2}$$

such that the HF self-consistency equations are obeyed (see Supplementary Note A4) and the Hamiltonian assumes a diagonal quadratic form within the mean-field approximation, i.e.,

$$\hat{H} = \sum_{\boldsymbol{k},p} \epsilon_{\boldsymbol{k},p} \bar{d}^\dagger_{\boldsymbol{k},p} \bar{d}_{\boldsymbol{k},p} + \ldots, \tag{3}$$

where the ellipsis indicates terms beyond mean-field. The transformations in Equation (2) are obtained in an iterative approach until a specified tolerance is reached.

The ground state within HF is given by filling the lowest fermionic states in Equation (3), which we will use as our RS, denoted by $|RS\rangle$ in the following. Importantly, though, HF also defines an entire basis via Equation (2), which is approximately related to the spectrum of the full Hamiltonian and parametrized by $\epsilon_{\boldsymbol{k},p}$. We leverage both the spectrum $\epsilon_{\boldsymbol{k},p}$ and its associated basis to improve sampling efficiency and physical interpretability within the VMC framework. As summarized graphically in Fig. 1(a, b), we express the many-body state in the HF basis (2) and denote the associated computational basis by $|\boldsymbol{s}\rangle$, where $\boldsymbol{s} = \left( s_1, \ldots, s_{N_k} \right)$ labels the occupations of the fermionic modes created by $\bar{d}^\dagger_{\boldsymbol{k},p}$ in the $N_k$ different electronic momenta $\boldsymbol{k}$. Our variational many-body ansatz then reads as

$$\left|\Psi_{\{\boldsymbol{\theta},\alpha\}}\right\rangle = \alpha|RS\rangle + \sqrt{1-\alpha^2} \sum_{\boldsymbol{s} \neq RS} \psi_{\boldsymbol{\theta}}(\boldsymbol{s})|\boldsymbol{s}\rangle, \tag{4}$$

where $\psi_{\boldsymbol{\theta}}(\boldsymbol{s}) \in \mathbb{C}$ is a neural network representation[1] of the amplitudes for the states $\boldsymbol{s}$ that are not the RS, and $\alpha$ is an additional variational parameter describing the weight associated with the RS. Note that a global phase choice allows us to take $\alpha \in \mathbb{R}$ without loss of generality.

The motivation for the variational parameter $\alpha$ is two-fold. First, it explicitly quantifies deviations of the ground state from the RS, which for HF refers to the optimal product state. A ground state being close to the RS is then reflected by $\alpha$ approaching unity, while small $\alpha$ will indicate strong deviations from a product state. As such, our approach combines the interpretability of HF with the lack of being constrained to (the vicinity of) a Slater determinant. We emphasize that different HF calculations, e.g., restricted to be in certain symmetry channels, can be used and compared. Secondly, through Equation (4), the NQS can

solely focus on the corrections $\delta E$ to the RS energy $E_{RS}$. Since the RS is never sampled by the NQS by construction, this separation is beneficial when HF captures the dominant ground state contributions, as targeting corrections would be hindered by low acceptance probabilities in Metropolis-Hastings sampling[40]—a phenomenon analogous to mode collapse in generative adversarial networks[41,42]. If HF is not a good approximation, there is, in general, no reason why splitting up the contribution of the RS would be detrimental to the network's performance.

It remains to discuss how the other states, $\boldsymbol{s} \neq RS$, are described through $\psi_{\boldsymbol{\theta}}(\boldsymbol{s})$ which depends on a set of parameters $\boldsymbol{\theta} \in \mathbb{R}^n$. These parameters are jointly optimized with $\alpha$ according to

$$\arg\min_{\{\boldsymbol{\theta},\alpha\}} E(\boldsymbol{\theta}, \alpha) = \arg\min_{\{\boldsymbol{\theta},\alpha\}} \frac{\left\langle \Psi_{\{\boldsymbol{\theta},\alpha\}} \middle| \hat{H} \middle| \Psi_{\{\boldsymbol{\theta},\alpha\}} \right\rangle}{\left\langle \Psi_{\{\boldsymbol{\theta},\alpha\}} \middle| \Psi_{\{\boldsymbol{\theta},\alpha\}} \right\rangle}, \tag{5}$$

i.e., via a minimization of the energy functional $E(\boldsymbol{\theta}, \alpha)$ (see Methods section). We emphasize that this approach is distinct from neural network backflow[43,44], but not mutually exclusive, as we use the HF basis to express the many-body state rather than dressing its single-particle orbitals with many-body correlations. While other approaches are feasible, too, we here employ a Transformer[21,26] to represent the Born distribution $q_{\boldsymbol{\theta}}(\boldsymbol{s}) = |\psi_{\boldsymbol{\theta}}(\boldsymbol{s})|^2 / \sum_{\boldsymbol{s}'} |\psi_{\boldsymbol{\theta}}(\boldsymbol{s}')|^2$ autoregressively, i.e.,

$$q_{\boldsymbol{\theta}}(\boldsymbol{s}) = \prod_{i=1}^{N_k} q\left(s_i | s_{i-1}, \ldots, s_1\right). \tag{6}$$

From this distribution, both the amplitudes and phases are obtained for the associated wave functions, $\psi_{\boldsymbol{\theta}}(\boldsymbol{s}) = \sqrt{q_{\boldsymbol{\theta}}(\boldsymbol{s})} e^{i\phi_{\boldsymbol{\theta}}(\boldsymbol{s})}$, from the Transformer's latent space (see Fig. 1b). Both components are calculated from the same output of the final Addition and Normalization layer of the Transformer. The amplitude is obtained through an affine linear transformation followed by a softmax activation function, while the phase uses a scaled softsign activation function to ensure $\phi_{\boldsymbol{\theta}}(\boldsymbol{s}) \in [-\pi, \pi]$[23,26]. This approach guarantees that the output of the Transformer output yields normalized conditional probabilities in Equation (6)[18].

## Model Hamiltonian

To test and explicitly demonstrate our methodology, we construct a concrete minimal model of the form given in Equation (1). The model has exact strong and weak coupling limits that can be used as effective theories $\hat{H}_0$—together with HF—for intermediate coupling regimes.

It describes spinless, one-dimensional electrons which can occupy two different bands, $a = \pm$, as described by the creation and annihilation operators $d^\dagger_{k,a}$ and $d_{k,a}$, respectively. They interact through a repulsive Coulomb potential $V(q) = (2N_k(1+q^2))^{-1}$. More explicitly, the Hamiltonian reads as

$$\hat{H} = t \sum_{k \in BZ} \cos(k) d^\dagger_k \sigma_z d_k + U \sum_{q \in \mathbb{R}} V(q) \rho_q \rho_{-q}, \tag{7}$$

where the momenta $k$ are defined on the first Brillouin zone $(BZ) := [-\pi, \ldots, \pi - 2\pi/N_k]$ of a finite system with $N_k$ sites and $\sigma_j$ $(j = 0, x, y, z)$ are the Pauli matrices in band space. The density operator is given by

$$\rho_q = \sum_{k \in BZ} \left( d^\dagger_{k+q} \mathcal{F}(k,q) d_k - \sum_{G \in RL} \delta_{q,G} f_1(k,G) \right), \tag{8}$$

where $RL = 2\pi\mathbb{Z}$ is the reciprocal lattice and the "form factors" read as $\mathcal{F}(k,q) = f_1(k,q) + i\sigma_y f_2(k,q)$; for concreteness, we choose $f_1(k,q) = 1$ and $f_2(k,q) = 0.9\sin(k)(\sin(q) + \sin(k+q))$ in our computations below.

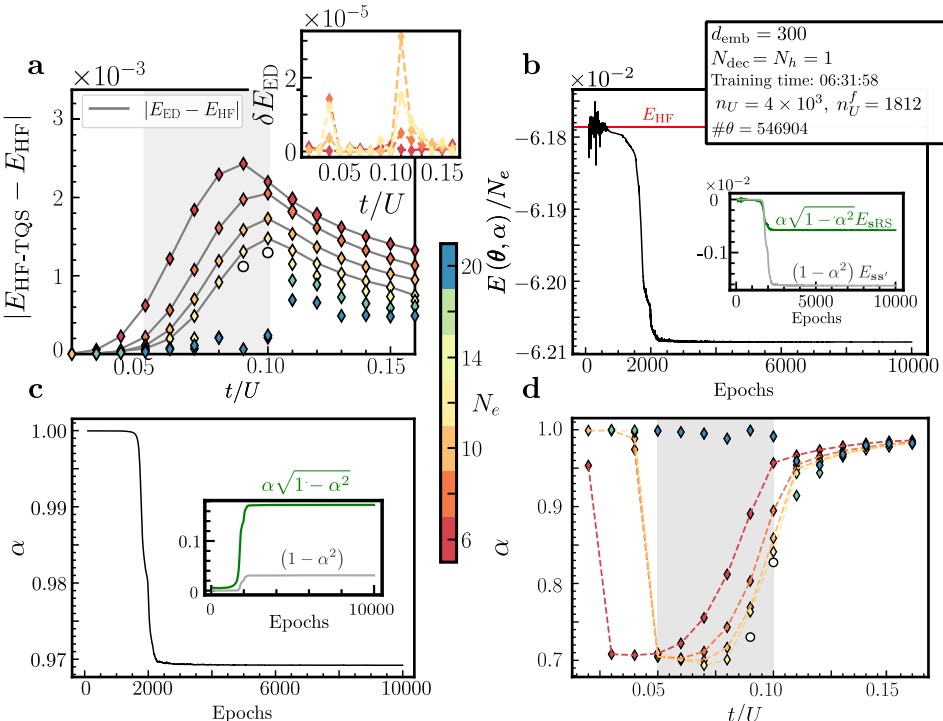

**Fig. 2 | Performance of HF-TQS for different system sizes and couplings $t/U$.**
**a** Difference between the HF-TQS ground state energy per electron and HF as a function of $t/U$ at various system sizes $N_e$. The solid lines show the difference between ED and the HF ground state energy. The inset shows the absolute value of the relative error $\delta E_{\mathrm{ED}} = |E_{\mathrm{HF-TQS}} - E_{\mathrm{ED}}|$. The corresponding converged $\alpha$ values [according to Equation (14)] are shown in panel **c**. The gray regions indicate the vicinity of the metal-insulator transition. We fix $n_U = 4 \times 10^3$ in this region to highlight how this parameter controls the accessible corrections. Therefore, the break in trend for the corrections at $N_e \geq 14$ highlights that a larger $n_U$ is necessary to

correctly capture them. To illustrate this point, the white circles in panels (**a**, **d**) for $N_e = 14$ were computed using $n_U = 17{,}000$. We refer the reader to the main text for more details. **b** Convergence of the ground state energy per electron and of $\alpha$ (panel **c**) during training for $t/U = 0.16$ and $N_e = 30$. The total number of unique states $n_U^f$ indicates how many states are retained by the Transformer from the initial value $n_U$ determined in Fig. 1c. Training was performed on one NVIDIA H100 GPU with the displayed network hyperparameters as defined in Fig. 1b (see also Supplementary Note B6). The total number of network parameters is denoted by $\#\boldsymbol{\theta}$.

Note that this model is non-sparse since all momenta are coupled and, as such, is generally expected to be challenging to solve. It is inspired by models of correlated moiré superlattices, most notably of graphene, which exhibit multiple low-energy bands that are topologically obstructed[45,46]; they can, hence, not be written as symmetric local theories in real space and are, thus, typically studied in momentum space[47–49].

Furthermore, the strong coupling limit, $t/U \to 0$, of Equation (7) can be readily solved: to this end, we introduce a new basis defined by $U_k = \begin{pmatrix} 1 & -i \\ 1 & i \end{pmatrix} / \sqrt{2}$ in Equation (2) which diagonalizes the form factors $\mathcal{F}(k, q)$ at all momenta. It follows (see Supplementary Note A1) that, at half-filling (the number of electrons $N_e = N_k$), any of the states $| \pm \rangle = \prod_k \bar{d}_{k, \pm}^\dagger |0\rangle$ are exact ground states in the limit $t/U \to 0$; which of the two ground states is picked is determined by spontaneous symmetry breaking: the Hamiltonian is invariant under the anti-unitary operator $PT$ with action $PT \bar{d}_{k, \pm} (PT)^\dagger = \bar{d}_{k, \mp}$, which is broken by both of these states. The resulting symmetry-broken phase can be shown to exhibit a finite gap. Importantly, this strong coupling limit defines another natural computational basis and associated $|RS\rangle = |+\rangle$ or $|-\rangle$, which we will use and compare with the HF basis defined in the previous section; in analogy to twisted bilayer graphene[48], we will refer to this strong-coupling basis as "chiral basis".

In contrast, at large $t/U$, the non-interacting term in Equation (7) dominates and we obtain a symmetry-unbroken metallic phase. As such, there is an interaction-driven metal-insulator transition at half-filling at some intermediate value of $t/U$ ($\simeq 0.14$ according to HF). To

be able to compare both chiral and HF bases and since half-filling has the largest Hilbert space, we will focus on $N_e = N_k$ in the following. Furthermore, we will neglect double-occupancy of each of the $N_e$ momenta for simplicity such that the basis states $|\boldsymbol{s}\rangle$, with $s_k \in \{0, 1\}$, in Equation (4) can be compactly written as $|s\rangle = \prod_{k=1}^{N_k} \bar{d}_{k, (-1)^{s_k}}^\dagger |0\rangle$.

## Hartree-Fock as an effective theory

We first discuss the results using HF as $\hat{H}_0$. The solid gray lines in Fig. 2a show the deviations of the HF ground state energy from that obtained by ED for system sizes $N_e$ where the latter is feasible. As expected, the corrections exhibit a higher magnitude near the metal-insulator transition (gray region). In the metallic regime ($t/U > 0.10$), the corrections decay more gradually, forming an extended tail. In contrast, in the insulating regime ($t/U < 0.05$), the corrections decrease rapidly as $N_e$ increases. To simultaneously display the performance of the Transformer-corrected ansatz (4) using the HF basis—which we refer to as HF-TQS from now on—the colored markers in Fig. 2a show the deviations of HF from the HF-TQS ground-state energy. The fact that they are very close to the deviation of HF to ED for all parameters demonstrates the expressivity and convergence of our approach; this can also be more explicitly seen in the inset that directly shows the difference in ground-state energy between ED and HF-TQS.

Naturally, the HF-TQS ansatz can also be applied to larger system sizes not accessible in ED. For instance, in Fig. 2b–c, we show the variational energy and $\alpha$ during training for $N_e = 30$ electrons at $t/U = 0.16$. Here and similarly on the low-$t/U$ side of the phase transition, we obtain fast convergence and systematic corrections to the HF energy consistent with the trend at smaller $N_e$ in Fig. 2a, although we

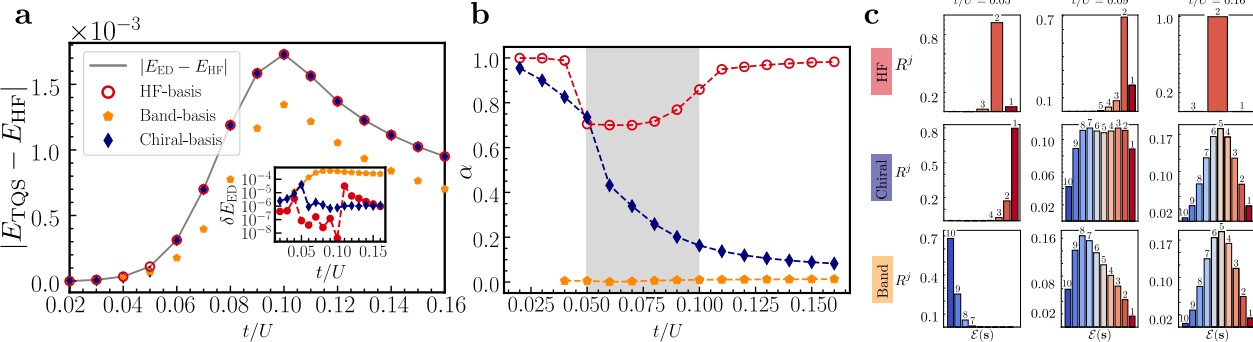

**Fig. 3 | Performance comparison of the TQS with distinct effective theories.**
**a** Difference between the TQS (with distinct effective theories labeled by the markers) ground state energy per electron and HF as a function of $t/U$ for $N_e = 10$. The solid line shows the difference between ED and the HF ground state energy. The inset shows the absolute value of the relative error $\delta E_{ED} = |E_{TQS} - E_{ED}|$ on a log scale. **b** Converged $\alpha$ for the TQS (markers) in panel **a** as a function of $t/U$, with dashed lines as a guide to the eye. The gray region indicates the vicinity of the metal-insulator transition. **c** Histograms showing the total relative frequencies $R^j$, according to Equation (10), for the excitation classes $\mathcal{E}(s)$ from Equation (9). These quantities represent the importance of corrections for each particle-hole excitation class. From left to right, the columns correspond to $t/U$ values in the insulating, critical, and metallic regimes, respectively.

just use the moderately large number of $n_U = 4 \times 10^3$ unique samples (cf. Fig. 1c). The data in Fig. 2b–c reveals that the Transformer properly captures the non-product corrections to the HF state. In fact, we can see that the converged Transformer only ends up having to sample $n_u^f = 1812$ distinct states (out of the $\simeq 10^9$ total states). This efficiency extends to even larger systems, as we demonstrate in Supplementary Note B7 with results up to $N_e = 60$.

The situation is different in the critical region, where the method's performance is primarily constrained by our current choice of a comparatively small total number of uniquely sampled states $n_U$. This leads to the drop (increase) of the energy correction ($\alpha$) in Fig. 2a, d for large system sizes in the gray region. Here a larger number of states is required for an accurate representation. When $n_U$ is sufficiently large to represent a substantial portion—or even the entirety—of the Hilbert space $\mathcal{H}$, the HF-TQS converges with good accuracy in this region. As we will see later, all effective theories exhibit the same behavior in the gray region, confirming this is a challenging regime for the Transformer-NQS to solve, regardless of the effective theory considered in this work. The trend change observed for the corrections in this region in Fig. 2a then comes naturally from the fact that we have fixed $n_U = 4 \times 10^3$ for all system sizes $N_e$, which seems insufficient for $N_e \geq 14$. To demonstrate this, we increased $n_U$ leading to the white circles in Fig. 2a, d (see Supplementary Note B7 for more details).

The relevance of the RS can also be conveniently seen from the parameter $\alpha$ in Fig. 2b. Away from the critical region, $\alpha$ approaches 1 signaling that HF becomes an increasingly accurate approximation while, within the critical region, we find $\alpha \simeq \sqrt{1 - \alpha^2} \simeq 0.7 \simeq 1/\sqrt{2}$, indicating that only about half of the ground state or half of its energy [cf. Equation (14)] is described by the HF state.

Finally, we point out that the learning rate $\lambda_{\alpha_0}$ for optimizing $\alpha_0$ was set to a fixed value, such that the training dynamics is dominated by the one of $\boldsymbol{\theta}$ (Fig. 2d). While alternative learning rate scheduling strategies could be proposed, they should be done with care. Specifically, we observed that low values of $\lambda_{\alpha_0}$ can cause the optimization of $\boldsymbol{\theta}$, according to Equation (15), to become trapped in local minima, particularly near the phase transition.

## Other effective theories

We next compare the performance when using the HF basis with that of the chiral basis, whose associated RS is expected to provide a good approximation to the ground state for small $t/U$. To this end, we show in Fig. 3a the deviation of the variational ground state energy from HF (main panel) and ED (inset) for these bases choices. We see that the chiral and HF bases both provide accurate representations of the ground state across the entire phase diagram demonstrating again that

the method is not intrinsically biased to being close to the RS, which, for the chiral basis, is not a good approximation for the ground state away from $t/U \to 0$; this is also confirmed by the behavior of the respective $\alpha$ shown in Fig. 3b: for the HF basis, it only dips significantly below 1 in the critical region, where non-product-state corrections are crucial, while dropping to zero for increasing $t/U$ in the chiral basis.

To analyze the performance of our ansatz further, in Fig. 3a, b, we also show results using the band basis [$U_k = \mathbb{1}$ in Equation (2)], and choose a fully filled band (e.g., $a = -$) as RS which, importantly, is not close to the ground state for any $t/U$—not even in the non-interacting limit [as can be seen in Equation (7), the band occupation has to change with momentum for $U = 0$]. In line with these expectations, we find $\alpha \ll 1$ in the entire phase diagram, see yellow pentagon markers in Fig. 3(b). As expected from Equation (4), the formalism then reduces to standard Transformer-NQS approaches in this regime. Nonetheless, the expressivity of the Transformer in the ansatz (4) allows to approximate the ground-state energy better than HF; it is not quite as good as in the HF or chiral basis which seems natural since the RS does not have any simple relation to the ground state in any part of the phase diagram. Thus, representing it and sampling from it is generically expected to be more challenging than in physics-informed bases. We checked that, for larger $t/U$, the transformer converges to the exact ground state energy also in the band basis as the asymptotic ground state is just one of the basis states (see Supplementary Fig. 5).

Additional important details about the wavefunction and sampling efficiency in the different bases can be revealed by studying the contributions of the various basis states. To group them, we recall that each $|s\rangle$ in Equation (4) is labeled by $\boldsymbol{s} = (s_1, \dots s_{N_k})$, $s_k \in \{0, 1\}$, and with the convention $|RS\rangle = |(1, 1, \dots, 1)\rangle$ it makes sense to use the number of "excitations" or "flips"

$$\mathcal{E}(\boldsymbol{s}) := \sum_{k=1}^{N_k} (1 - s_k) \quad (9)$$

relative to the RS; in the case of the HF basis, these are in one-to-one correspondence to the particle-hole pairs described by the mean-field Hamiltonian (3). For $N_e = 6$, for example, states like $|111110\rangle$ and $|111101\rangle$ belong to the class with $\mathcal{E} = 1$, i.e., with a single excitation above the RS. We also define

$$R^j := \sum_{\boldsymbol{s}|\mathcal{E}(\boldsymbol{s})=j} r(\boldsymbol{s}) \quad \text{for} \quad j = 1, \dots N_e, \quad (10)$$

where $r(\boldsymbol{s})$ are the relative frequencies defined in Equation (12). This quantity represents the relative weight of the ground state

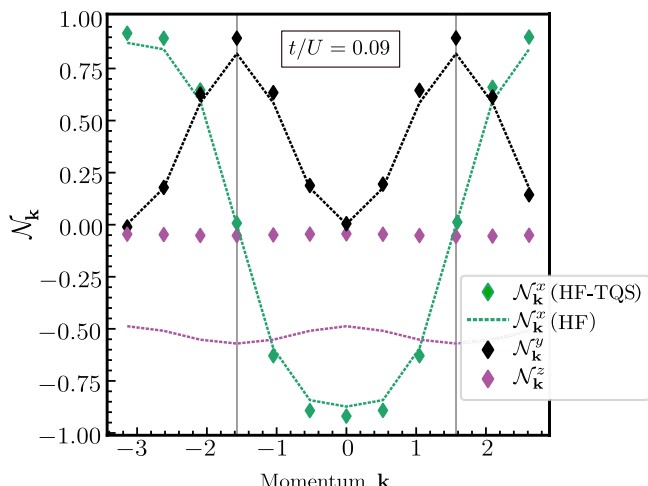

**Fig. 4 | Momentum-resolved fermionic bilinears $\mathcal{N}_{\boldsymbol{k}}^{j}$.** HF-TQS results (markers) as a function of momentum $\boldsymbol{k}$ for the observable defined in Equation (11), in comparison to those obtained solely from HF (dashed lines) for $N_e = 12$ at $t/U = 0.09$.

wavefunction in the sector with $j$ excitations, normalized such that $\sum_{j=1}^{N_e} R^j = 1$ (excluding the reference state with $j = 0$). More formally, if we define the projector $\hat{P}_j = \sum_{\boldsymbol{s}} \delta_{\mathcal{E}(\boldsymbol{s}),j} |\boldsymbol{s}\rangle \langle \boldsymbol{s}|$ onto the subspace with $\mathcal{E}(\boldsymbol{s}) = j$, then $R^j = ||\hat{P}_j|\Psi\rangle||^2 / (1 - \alpha^2)$, where $|\Psi\rangle$ is the ground state. These quantities provide a quantitative measure of which particle-hole excitation sectors contribute most to the corrections needed to reach the true ground state from $\hat{H}_0$.

The histograms in Fig. 3c show these quantities for the three respective values of $t/U$ indicated in Fig. 3a. The chiral and band bases are fundamentally limited by the curse of dimensionality: the ground state physics cannot be captured by just a few dominant basis states, as demonstrated by the significant contributions of all $R^j$ away from the insulating regime. This broad distribution would consequently limit their applicability for larger system sizes $N_e$. In contrast, the ground state representation in the HF basis for both the insulating and metallic parameter range is dominated by low-order excitations relative to the HF state, as expected from Equation (3). This behavior enables accurate calculations for $N_e \geq 16$ in Fig. 2, both in the metallic and insulating regimes, in spite of the non-sparse nature of the Hamiltonian.

Unlike coupled cluster methods in quantum chemistry, for example, the Transformer independently selects the most important excitation classes. This can be seen particularly from the HF-basis histogram close to the phase transition ($t/U = 0.09$), as an increasing number of higher order excitations starts contributing to the ground-state energy. The number of accessible classes is then limited by only two factors: the total number of unique partial strings $n_U$ allowed in the batch-autoregressive sampler (Fig. 1c), and the Transformer's expressiveness, which is primarily controlled by $d_{emb}$, $N_h$ and $N_{dec}$[50] (see Fig. 1b and Supplementary Fig. 2). Interestingly, though, we see in Fig. 3c that even close to the phase transition, the HF basis clearly benefits more from importance sampling than the other bases.

### Observables

Apart from the ground-state energy of Equation (7), we can naturally estimate other observables, such as the momentum-resolved fermionic bilinears,

$$\mathcal{N}_{\boldsymbol{k}}^{j} = \bar{d}_{\boldsymbol{k}}^{\dagger} \sigma_j \bar{d}_{\boldsymbol{k}}, \quad j = x, y, z, \qquad (11)$$

where $\bar{d}_{\boldsymbol{k}}$ are the fermionic operators in the chiral basis. In Fig. 4, we show their expectation values within HF and HF-TQS in the critical region ($t/U = 0.09$). As the dispersion involves [first term in Equation (7)] $\sigma_x$ in the chiral basis, it is natural to recover the cos-like shape in

$\langle \mathcal{N}_{\boldsymbol{k}}^{x} \rangle$. Most importantly, $\langle \mathcal{N}_{\boldsymbol{k}}^{z} \rangle$, which describes the symmetry breaking in the insulating regime, is sizeable in HF, showing that the system is already in the symmetry-broken, insulating regime. However, the additional quantum corrections from our HF-TQS approach lead to a much smaller almost vanishing $\langle \mathcal{N}_{\boldsymbol{k}}^{z} \rangle \simeq 0$. This is in line with general expectations that HF overestimates the tendency to order. Moreover, corrections to $\mathcal{N}_{\boldsymbol{k}}^{y}$ are more pronounced near the points where the kinetic term in Equation (7) changes sign (vertical gray lines in the plot). In combination with the fact that the deviations between HF and HF-TQS are much less pronounced away from the critical region (see Supplementary Fig. 4), these results demonstrate that the value of the parameter $\alpha$ ( $\simeq 0.76$ at $t/U = 0.09$) also serves as an indicator of expected deviations from HF predictions for other physical observables.

### Hidden representation

Finally, we investigate the influence of the three different bases on the Transformer's latent space by projecting the high-dimensional parametrization of $q_{\boldsymbol{\theta}}(\boldsymbol{s})$ onto low-dimensional spaces using principal component analysis (PCA)[51–53]. We apply this method to the set of vectors $\{\boldsymbol{H}(\boldsymbol{s}) = \sum_{j}^{N_e} \boldsymbol{h}_j(\boldsymbol{s}) | \forall \boldsymbol{s} \neq RS\}$[28], which are obtained at the output of the Transformer's $N_{dec}$ layers (see Fig. 1b). For visual clarity, we focus on $N_e = 10$ electrons. Figure 5 shows the first and second principal components of PCA for all bases at $t/U = 0.04$ (insulator), $t/U = 0.12$ (close to critical region) and $t/U = 0.16$ (metal). To first compare the two natural, energetically-motivated bases—the HF and chiral basis— we see that the states are indeed approximately ordered based on the classes defined via Equation (9) in the regimes where they are expected to be natural choices, i.e., for all $t/U$ (small $t/U$) for the HF (chiral) basis. This illustrates that the physical motivation for choosing these respective bases is not only visible in the histograms in Fig. 3b and the sampling efficiency but also "learned" by the Transformer's hidden representation. While some clear structure also emerges for the band basis, we emphasize that the labels $\mathcal{E}(\boldsymbol{s})$ do not directly translate to the energetics of the states: as discussed above, the RS is never close to the ground states in any regime, such that the number of excitations $\mathcal{E}$ above it also does not present clear energetic relevance either. Only for large $t/U$ does a related quantity, the excitations away from the product ground state, that can be defined in this basis become relevant. The Transformer appears unable to uncover any additional emergent structure, which is likely related to the poor performance of the band basis, as shown in Fig. 3a. Hence, interpretability of these structures is not automatically ensured, as the above example illustrates.

### Discussion

We have introduced and demonstrated a modified transformer-based variational description of the ground state of a many-body Hamiltonian, which is based on first choosing an energetically motivated basis $\{|RS\rangle, |\boldsymbol{s}\rangle\}$, according to Equation (4) and Fig. 1a. We showed that HF provides a very natural and general route towards finding such a basis since the associated mean-field Hamiltonian (3) encodes an approximate energetic hierarchy of the states. As a second example, we used a basis defined in the strong-coupling limit. Overall, our approach has the following advantages: (i) there is a single parameter, $\alpha$, which quantifies how close the (variational representation of the) ground state is to $|RS\rangle$; for instance, for the HF basis, this would be the mean-field-theory prediction, i.e., the Slater determinant closest to the true ground state; (ii) except for right at the critical point, the HF basis is found to be particularly useful for improving the sampling efficiency since only a small subset of the exponentially large basis states contribute. This is expected based on general energetic reasoning and is most directly visible in the histograms in Fig. 3c. Finally, (iii) the physical nature of these bases also allows for a clear interpretation of the different contributions, e.g., as excitations on top of the RS, which we also recover in the transformer's hidden representation (see Fig. 5).

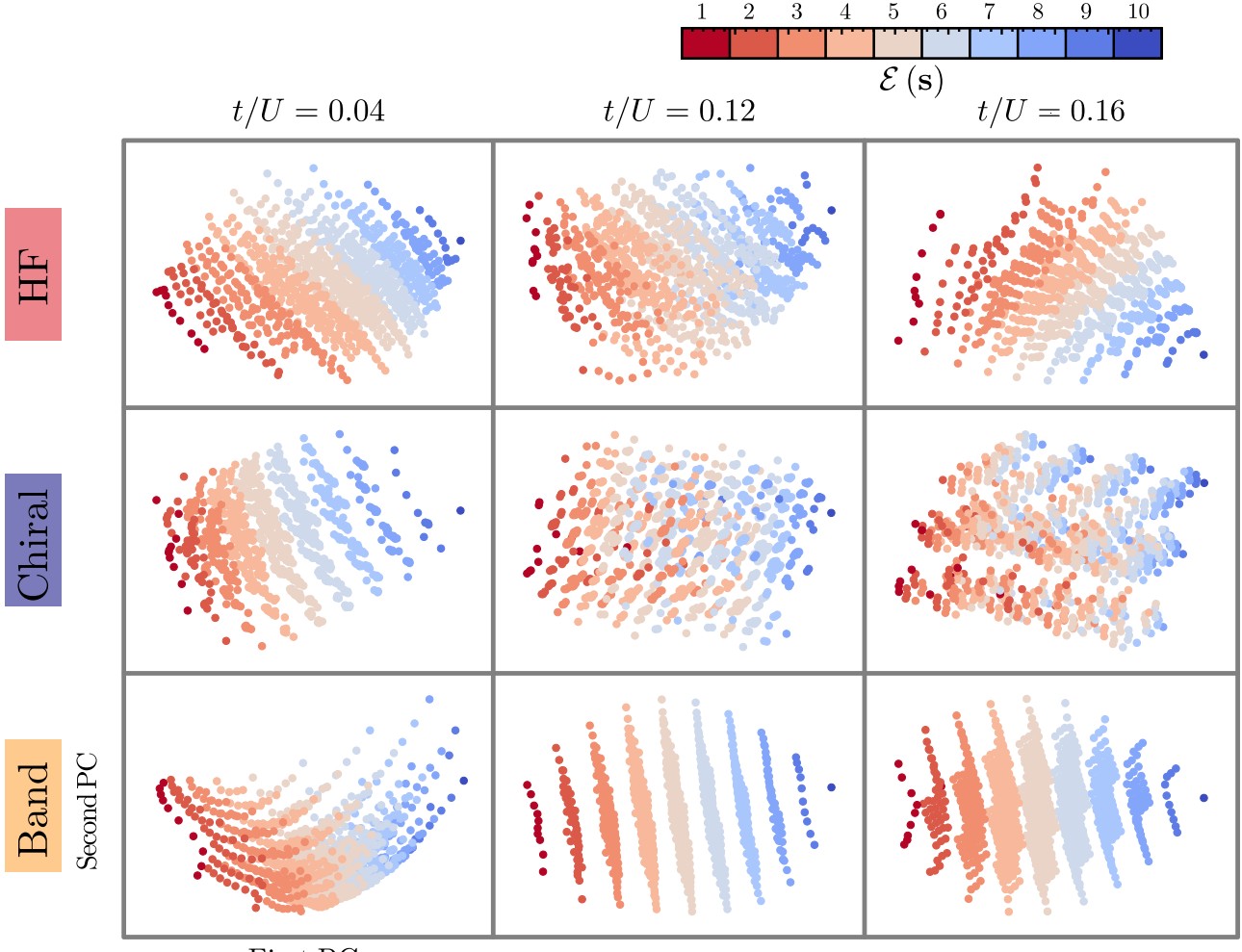

**Fig. 5 | Visualization of the Transformer's latent space.** Results are shown for $N_e = 10$ electrons at different values of $t/U$ for the band, chiral and HF bases. Each point represents a basis state $\boldsymbol{s}$, which is colored according to the class label $\mathcal{E}(\boldsymbol{s})$ [cf. Equation (9)], and has been obtained by projecting the respective latent space features $\boldsymbol{H}(\boldsymbol{s})$ onto the first two principal components (PCs) using PCA. All simulations use embedding dimension $d_{emb} = 300$ with single attention head and decoder layer ($N_h = N_{dec} = 1$).

Several directions can be addressed in future work. First, applying this methodology to different Hamiltonians and deep learning architectures is a natural next step to determine more generally under which conditions only a small subset of basis states is required for the ground state in metallic and insulating regimes. In particular, since the mean-field approximation becomes more accurate in higher dimensions, one would expect HF to provide even greater advantages as an effective theory in higher dimensions. Additionally, inspired by Ref. 36, where orbital rotations applied to determinant-based wavefunctions[43,44,54] were shown to improve variational energies and to modify orbitals in certain scenarios, it would be interesting to investigate whether effective theories could provide similar sampling benefits in the context of such ansätze.

From a methodological perspective, efficiency improvements could be achieved through the usage of modified stochastic reconfiguration techniques for the optimization of the network parameters[10,11], and with the incorporation of symmetries in the HF-based ansatz[55,56]. For systems with non-sparse Hamiltonians like our current model, the implementation of the recently proposed GPU-optimized batch auto-regressive sampling without replacement[57] should also be beneficial. Furthermore, our approach could be used to test the validity and accuracy of different effective theories by using them as $\hat{H}_0$ in Fig. 1a to define the computational basis.

## Methods

### Local energy estimators

The energy expectation values for the corrections $\delta E$ are calculated as a weighted average over a set $\mathcal{S}$ of $n_U$ unique states (from a batch of $N_s$ sampled states $\boldsymbol{s}$) from $q_{\boldsymbol{\theta}}(\boldsymbol{s})$ through the batch auto-regressive sampler[57–59] (see Fig. 1c) as

$$\langle \delta E \rangle = \mathbb{E}_{\boldsymbol{s} \sim q_{\boldsymbol{\theta}}} \left[ H_{\text{loc}}(\boldsymbol{s}) \right] \simeq \sum_{\boldsymbol{s} \in \mathcal{S} \neq \text{RS}} H_{\text{loc}}(\boldsymbol{s}) r(\boldsymbol{s}). \tag{12}$$

Here, $r(\boldsymbol{s}) = n(\boldsymbol{s})/N_s$ represents the relative frequency of each state $\boldsymbol{s}$ and

$$H_{\text{loc}}(\boldsymbol{s}) = \sum_{\boldsymbol{s}' \neq \text{RS}} \frac{\langle \boldsymbol{s}|\hat{H}|\boldsymbol{s}'\rangle \psi_{\boldsymbol{\theta}}(\boldsymbol{s}')}{\psi_{\boldsymbol{\theta}}(\boldsymbol{s})} \tag{13}$$

are the typical local estimators. According to Equation (4), the energy functional is divided into sectors

$$E(\boldsymbol{\theta}, \alpha) = \alpha^2 E_{\text{RS}} + (1 - \alpha^2) \mathbb{E}_{\boldsymbol{s} \sim q_{\boldsymbol{\theta}}} \left[ H_{\text{loc}}(\boldsymbol{s}) \right] + \\ + 2\alpha\sqrt{1 - \alpha^2} \text{Re}\left( \mathbb{E}_{\boldsymbol{s} \sim q_{\boldsymbol{\theta}}} \left[ H_{\text{loc}}^{\text{RS}}(\boldsymbol{s}) \right] \right), \tag{14}$$

with the modified local estimator $H_{\text{loc}}^{\text{RS}}(\boldsymbol{s}) = \langle \boldsymbol{s}|\hat{H}|\text{RS}\rangle/\psi_{\boldsymbol{\theta}}(\boldsymbol{s})$. The network parameters $\boldsymbol{\theta}$ are optimized as usual with the gradients of the

expression (14) given by

$$\nabla_{\boldsymbol{\theta}} E(\boldsymbol{\theta}, \alpha) = 2\mathrm{Re}\left(\mathbb{E}_{\boldsymbol{s} \sim q_{\boldsymbol{\theta}}}\left[\mathcal{H}_{\mathrm{loc}}(\boldsymbol{s}, \alpha) \cdot \nabla_{\boldsymbol{\theta}} \log \psi_{\boldsymbol{\theta}}^*(\boldsymbol{s})\right]\right), \qquad (15)$$

with

$$\mathcal{H}_{loc}(\boldsymbol{s}, \alpha) = \left(\left(1 - \alpha^2\right) H_{\mathrm{loc}}(\boldsymbol{s}) + \alpha\sqrt{1 - \alpha^2} H_{\mathrm{loc}}^{\mathrm{RS}}(\boldsymbol{s})\right).$$

To prevent numerical instabilities during the optimization of Equation (4), it is necessary to constrain $\alpha$ with the parametrization $\alpha = (1 + \tanh \alpha_0)/2$ to the interval $[-1, 1]$. After updating the network parameters $\boldsymbol{\theta}$ at each iteration, the reweighting parameters are dynamically adjusted according to the gradient of $E(\boldsymbol{\theta}, \alpha)$ in Equation (14) with respect to $\alpha_0$, i.e.,

$$\nabla_{\alpha_0} E(\boldsymbol{\theta}, \alpha) = 2\alpha \nabla_{\alpha_0} \alpha \left[E_{\mathrm{RS}} - E_{\boldsymbol{ss'}} + \frac{E_{\boldsymbol{s}\mathrm{RS}}(1 - 2\alpha^2)}{2\alpha\sqrt{1 - \alpha^2}}\right], \qquad (16)$$

where $E_{\boldsymbol{ss'}} = \mathbb{E}_{\boldsymbol{s} \sim q_{\boldsymbol{\theta}}}\left[H_{\mathrm{loc}}(\boldsymbol{s})\right]$ and $E_{\boldsymbol{s}\mathrm{RS}} = 2\mathrm{Re}\left(\mathbb{E}_{\boldsymbol{s} \sim q_{\boldsymbol{\theta}}}\left[H_{\mathrm{loc}}^{\mathrm{RS}}(\boldsymbol{s})\right]\right)$. For the optimizer, we use stochastic gradient descent for Equation (16) and preconditioned gradient methods[60,61] for Equation (15) with adaptable learning rate schedulers (see Supplementary Note B6 for more details).

## Data availability
The minimal dataset required to reproduce the more data-intensive plots in Figs. 2a, d, 3a, b, and Supplementary Fig. 3 is available at https://doi.org/10.5281/zenodo.17600587[62]. Data for Figs. 2b, c, 4, 5, and remaining figures on the Supplementary Information can be readily reproduced using the provided source code.

## Code availability
The source code is publicly available at https://doi.org/10.5281/zenodo.17600587[62].

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

## Acknowledgements

M.S.S. thanks P. Wilhelm for discussions and previous collaborations. J.A.S. also acknowledges discussions with Y.-H. Zhang, S. Banerjee, L. Pupim, V. Dantas, P. Wilhelm, M. Mühlbauer, M. Medvidović, and J. Mögerle.

## Author contributions

Transformer simulations were performed by J.A.S., H.F. and exact diagonalization by M.P. and J.A.S. and analytical calculations on the SI by all authors. M.S.S. planned and supervised the project. All authors contributed to the writing of the manuscript.

## Funding

## Competing interests

The authors declare no competing interests.
