## [Transparent Peer Review file · Nature Communications]

Physics-informed Transformers for Electronic Quantum States

Corresponding Author: Mr João Augusto Sobral

Version 0:

Reviewer comments:

Reviewer #1

(Remarks to the Author)

This manuscript presents an interesting approach that integrates physics-informed basis selection with Transformer-based neural quantum states to improve the representation of many-body electronic quantum states. By leveraging a variational Monte Carlo framework with an autoregressive Transformer model, the authors aim to enhance both sampling efficiency and interpretability of neural quantum states. The proposed method is tested on a one-dimensional interacting fermionic model with a metal-insulator transition, demonstrating its effectiveness compared to other basis choices.

The study addresses important challenges in machine learning-based variational quantum state representations. While the introduction of a reference state and the comparative analysis of different reference bases provide novel insights, many of the techniques employed—such as autoregressive neural quantum states and variational Monte Carlo optimization—are already well established in the literature. Furthermore, prior studies have explored similar approaches that incorporate physics-inspired basis choices (e.g., Slater determinants) into neural quantum states. Given this, the manuscript may be better suited for a more topically focused journal.

I also have some specific questions and comments:

1. Near the metal-insulator transition, there appears to be a sudden qualitative change between $N_e = 12$ and $N_e = 14$. The results for $N_e = 14$ no longer follow the same trend as smaller system sizes, and $\alpha \approx 1$ instead of 0.7 near the transition. Could the authors verify whether this is a physical effect or a numerical artifact?

Additionally, ED results stop at $N_e = 12$, while the number of basis states sampled in the Transformer $n_U = 4000$ is approximately 2^{12} , which suggests that the algorithm may only work well when all basis states are explicitly covered. It would be beneficial to provide additional numerical results with larger n_U and, if feasible, ED results for slightly larger system sizes to confirm the trends.

2. For the Hamiltonian studied in this work, what is the previous state-of-the-art result? The current manuscript only compares TQS and ED results, but a lack of benchmarking against alternative numerical methods (e.g., other variational or quantum Monte Carlo approaches) makes it difficult to fully assess the contributions of this work. Including such comparisons would provide a clearer context for the significance of the results.

3. The largest system size considered is $N_e = 30$, which is reasonable for a proof-of-concept but leaves open questions regarding scalability. A discussion of how computational costs scale with increasing system size would provide a clearer picture of the practical feasibility of this approach for larger quantum systems.

4. The manuscript briefly describes the model architecture and optimization parameters, but additional details on how these were selected would improve clarity.

Minor comments: Some text in the figures is too small to be legible. The boxes in Fig. 3(a)(c) should be more clearly explained, and the scale of the y-axis in Fig. 3(c) is somewhat unclear.

(Remarks on code availability)

The code appears to be complete, but there is no instruction on how to run it. An updated README file with detailed instructions on how to reproduce the results in the main text would be helpful.

Reviewer #2

(Remarks to the Author)

In "Physics-Informed Transformers for Electronic Quantum States" by Sobral et al., the authors introduce a new approach to electronic quantum states search via innovative usage of autoregressive variational wave-functions. The presented method offers a degree of interpretability, which in particular presents a step forward. This work combines very well a reasonable level of physics insight, while still keeping the generic approach of neural network quantum states. The paper is well written and results are understandably presented. Overall, we are positive about this manuscript and would recommend it for publication pending the comments listed below.

1. Line 49: "TQS ... given their previously mentioned advantages"

Can the authors clarify these advantages and over what they are? It is stated that the transformer quantum states "have proven effective in providing highly accurate representation of ground states...", but it is not clear what the authors are really comparing the transformer to, here, and what evidence they have for these benefits.

2. Line 71-78: This is just a difficult sentence to read, could the authors split it and simplify?

3. Lines 182-189: Again, can this part be rewritten for clarity. Especially to better explain how the neural network is used to produce both outputs. This text alone is not entirely sufficient to make this point clear. The figure the text is referring to is very clear, so perhaps the solution is just to connect the two better.

4. Figures 2 and 3 are very dense as is the explanation of all the captions. Currently it takes much time and effort to understand what is plotted. We suggest creating more verbose captions to aid the understanding of the presented results.

5. Line 283: The slight inconsistency of nomenclature is a little confusing, the use of HF-TQS is valid and clear, but when this is used to refer to Figure 2, where this phrase is not used, it loses some of the clarity and makes it harder to easily flick from text to figure to see the point being made. Nonetheless, the result shown in Figure 2 is impressive.

6. Lines 290-296: The authors claim that this HF-TQS ansatz applies to larger systems, but this is not true for $t/U < 0.1$. This is addressed later in the text, but it seems odd not to mention it as a caveat when claiming that the method scales well.

7. Figure 3: We recommend a log scale for the inset of panel (a). It is clear that this inset shows that the band basis is much less performant, but it would be also interesting to see the relative performance of the other two bases shown, and this is not clear from the linear scale.

8. Lines 361-363: Please can the authors refer back to the evidence in their plots that support this claim, I think this would also help the reader switch from text to figures to convince themselves of the result.

9. Figure 4: The misalignment of the boxes is a little undesirable, it would make for a nicer figure to have consistent subplot size and placement.

10. Equation 15 and the surrounding text: the interpretation of R^i is a little unclear from first glance, a reader new to this topic would benefit from some interpretation of this quantity to give them something to compare to in their minds.

11. Lines 388-389: Why do the authors claim that this is the curse of dimensionality?

12. The code in the appended repository appears readable and complete and the key functions are sufficiently explained, but it is left to the user to navigate usage of this repository. We urge the authors to explain the file structure and example usage in the readme.

Overall, most of our comments are related to the presentation rather than the substance. The results are interesting and useful to the community. The work is creating a useful synergy between injection of some physics knowledge while still leveraging the flexibility of the physics-agnostic methods. After the comments above are addressed we will be pleased to recommend this manuscript for the publication.

(Remarks on code availability)
See the main report for feedback.

Reviewer #3

(Remarks to the Author)

(Remarks on code availability)

Reviewer #4

(Remarks to the Author)

(Remarks on code availability)

Version 1:

Reviewer comments:

Reviewer #1

(Remarks to the Author)

I appreciate the authors' detailed rebuttal and the substantial revisions they have already made. A few important issues still need attention before the manuscript is, in my view, ready for publication in Nature Communications.

1. The revised draft argues that the critical region is "intrinsically hard" and that the sample cap n_U must scale exponentially for HF-TQS to converge. That conclusion will be convincing only if the authors can show that no established algorithm performs well there. I recommend the authors include additional benchmarks obtained with a different state-of-the-art method (for example, tensor networks). Even a modest-size comparison would clarify whether the growth of n_U is a limitation of HF-TQS or a generic obstacle.

2. An adaptive or data-driven criterion for choosing n_U would be beneficial.

3. The present study focuses on a custom 1D, non-local momentum-space model inspired by moiré systems. Readers may wonder whether locality or dimensionality change the conclusions. I recommend including at least a brief test on a local lattice Hamiltonian (e.g. 1D Hubbard or J1-J2 chain) with established results.

I would be happy to recommend publication once these points are satisfactorily addressed.

(Remarks on code availability)

The revised github repo is clear and meets the standard of nature communications.

Reviewer #2

(Remarks to the Author)

Dear Editor,

we thank the authors for the revision of the manuscript. We are very happy with the implemented changes.

Re-reading the manuscript, we arrived at one more minor point:

figure 5: The authors claim that the structure of this latent space w.r.t $E(s)$ is an interpretable benefit. At the same time they discard the structure as not being able to be interpreted for the band basis. However, the structure is arguably the most clear in this case. The authors do suggest that

"While some clear structure also emerges for the band basis, we emphasize that the labels $E(s)$ do not directly translate to the energetics of the states: as discussed above, the RS is never close to the ground states in any regime, such that the number of excitations E above it also does not present clear energetic relevance either"

however (a) this downplays the amount of structure in the plot and (b) this only begs the question of if this latent space

structure is useful in the first place; if the structure remains despite $E(s)$ not being a relevant quantity, then how can we see this 'structure w.r.t $E(s)$ ' as being a robust interpretable quantity of interest.

It is of course not surprising and quite reasonable that the latent space cannot be interpreted in all instances, but we would still recommend adding one sentence along these lines to emphasize that the interpretability of the observed structure is not always guaranteed as underscored by the example above.

Provided this minor correction is addressed, we are happy to enthusiastically accept this manuscript for the publication in Nature Comm.

(Remarks on code availability)

Reviewer #3

(Remarks to the Author)

(Remarks on code availability)

Reviewer #4

(Remarks to the Author)

(Remarks on code availability)

Version 2:

Reviewer comments:

Reviewer #1

(Remarks to the Author)

I would like to thank the authors for their detailed responses and careful revisions. They addressed the main concerns raised in the previous round. I do not have any remaining concerns that would block publication. I recommend acceptance.

(Remarks on code availability)

Response to Reviewers NCOMMS-25-02094: “Physics-informed Transformers for Electronic Quantum States”

June 13, 2025

Responses to Report 1

Reviewer (1.1)

This manuscript presents an interesting approach that integrates physics-informed basis selection with Transformer-based neural quantum states to improve the representation of many-body electronic quantum states. By leveraging a variational Monte Carlo framework with an autoregressive Transformer model, the authors aim to enhance both sampling efficiency and interpretability of neural quantum states. The proposed method is tested on a one-dimensional interacting fermionic model with a metal-insulator transition, demonstrating its effectiveness compared to other basis choices. The study addresses important challenges in machine learning-based variational quantum state representations. While the introduction of a reference state and the comparative analysis of different reference bases provide novel insights, many of the techniques employed—such as autoregressive neural quantum states and variational Monte Carlo optimization—are already well established in the literature. Furthermore, prior studies have explored similar approaches that incorporate physics-inspired basis choices (e.g., Slater determinants) into neural quantum states. Given this, the manuscript may be better suited for a more topically focused journal.

Response

We thank the Reviewer for recognizing our approach’s novelty and potential insights. We clarify several key points and the work’s main contribution below:

- *“Furthermore, prior studies have explored similar approaches that incorporate physics-inspired basis choices (e.g., Slater determinants) into neural quantum states. Given this, the manuscript may be better suited for a more topically focused journal.”*

The sentence lacks supporting references. To the best of our knowledge, no similar work exists. The closest is Ref. [34] from our introduction, which shows that neural quantum states performance depends on the computational basis and suggests an optimal one may exist, though their focus was not interpretability.

For the part in the parenthesis, we suppose that the Reviewer is referring to neural network backflow (NNB) (Ref. [25] of the main text) or hidden fermion determinant states (HFDS) J. Robledo Moreno et al.: Proc. Natl. Acad. Sci. U.S.A. 119, 32

(2022). These are powerful *Ansätze* but fundamentally different since they restrict the wavefunction to determinant-based forms. We impose no such restriction, following Hibat-Allah et al.: Phys. Rev. Res. 2, 023358 (2020), for example. NNB and HFDS represent “restrictive biases” while our approach constitutes “soft inductive bias” according to the nomenclature in A. G. Wilson: arXiv:2503.02113 (2025), where bias refers to guidance on the hypothesis space. Given these distinctions and our results’ implications for both strongly correlated systems and quantum chemistry communities, this work belongs to a broad-scope journal like Nature Communications rather than a topically focused one.

- “...many of the techniques employed—such as autoregressive neural quantum states and variational Monte Carlo optimization—are already well established in the literature.”

Of course, we do not claim to introduce variational Monte Carlo nor autoregressive sampling and whenever previously well-established techniques are used, we acknowledge that in the main text. Instead, our main point is that combining these with an approximate effective theory (\hat{H}_0 in the main text) one can (i) improve sampling efficiency/numerical performance, (ii) yield more physically insightful/interpretable results, and (iii) our approach can be seen as a qualitative test of H_0 for a given problem. It should be clear from the derivations in the draft, based solely on the variational principle, that it is agnostic with respect to the architecture and *Ansätze* form. If one has a sufficiently close effective theory H_0 to the true ground state, one can potentially gain both interpretability (as defined on the main text) and computational efficiency, since fewer states may be needed to describe the ground state in certain regions of the phase diagram.

Reviewer (1.2)

For the Hamiltonian studied in this work, what is the previous state-of-the-art result? The current manuscript only compares TQS and ED results, but a lack of benchmarking against alternative numerical methods (e.g., other variational or quantum Monte Carlo approaches) makes it difficult to fully assess the contributions of this work. Including such comparisons would provide a clearer context for the significance of the results.

Response

About our model: We created this model from scratch specifically for this work and inspired by twisted van der Waals (vdW) materials. No other results exist besides those we present with HF, ED and analytically exact solvable limits. Our motivation was threefold:

- The vdW materials’ community makes extensive use of effective theories like Hartree-Fock, given the challenges of working with topologically obstructed models. Determining when these theories reliably describe the physics—and

how to go beyond them when they fail—is crucial;

- Since we have exact limits given by strong and weak couplings (small and large t/U), we can also test these different effective theories when going away from these limits;
- From a technical point of view and given the maturity of the neural quantum states’s field, we believe there is already substantial amount of work with local many-body models (some even exactly solvable for any combination of Hamiltonian coupling parameters). We believe it is also important (and timely, given the first point) to see how state-of-art architectures, like Transformers, perform with non-local models and to address challenges stemming from these scenarios.

About the significance of the results: We respectfully disagree that other numerical methods are necessary to validate the results already present in the manuscript. The three methods (HF, ED and analytical limits) employed here are sufficient for our goals, which were:

- To understand if and how effective theories \hat{H}_0 [identified in the main text as band, chiral and HF] can influence a neural network-based representation of the ground state of this model;
- To determine advantages of this approach, which we demonstrate in the metal and insulator regimes with HF for this specific model;
- To assess if one can recover any kind of interpretability from this, which stems from the considered effective theory.

In the revised version of the manuscript, we revised the motivation of our choice of Hamiltonian.

Reviewer (1.3)

Near the metal-insulator transition, there appears to be a sudden qualitative change between $N_e = 12$ and $N_e = 14$. The results for $N_e = 14$ no longer follow the same trend as smaller system sizes, and $\alpha \approx 1$ instead of 0.7 near the transition. Could the authors verify whether this is a physical effect or a numerical artifact? Additionally, ED results stop at $N_e = 12$, while the number of basis states sampled in the Transformer $n_U = 4000$ is approximately 2^{12} , which suggests that the algorithm may only work well when all basis states are explicitly covered. It would be beneficial to provide additional numerical results with larger n_U and, if feasible, ED results for slightly larger system sizes to confirm the trends.

Response

This was done deliberately for transparency, highlighting how performance around the critical region depends on n_U . We explained this in Section II.C in lines 304-312. All effective theories exhibit the same behavior, confirming this is an intrinsically

hard region to solve (see histograms in middle row of *Fig. 2c*).

Indeed, as pointed out by the referee, a larger n_U will be able to resolve this. Below we show an example with larger n_U for $N_e = 14$ at $t/U \approx 0.10$ to demonstrate this point. Since $n_U > 2^{14} > n_U^{\text{previous}} = 4000$, more corrections are found and α deviates more from unity close to the phase transition. Compare this histogram with the one at $t/U = 0.09$ in *Fig. 3c*. The final number of states n_U^f kept by the Transformer is actually very close to 2^{14} . However, the hierarchy over particle-hole excitation classes $\mathcal{E}(\mathbf{s})$ is maintained, contrasting with its absence in other effective theories (see *Fig. 3c*), with double particle-hole excitations still dominating the corrections. By consistency, similar results should apply for $N_e \geq 14$ in the gray region of *Fig. 2a*. Away from the critical regime, there is still substantial advantage since $n_U^f \ll 2^{N_e}$, for *any* system size N_e . We added a remark about this in the *Fig. 2* caption to avoid further confusion and included the results from *Fig. 1* in the supplement.

Figure 1: Convergence of the ground state energy per site for HF-TQS (a) and of α (b) as a function of epochs for $N_e = 14$ at $t/U = 0.10$ and $n_U = 17000$. (c) Histogram indicating the total relative frequencies R^j , according to Eq. (15) of the main text, for the excitation classes $\mathcal{E}(\mathbf{s})$ from Eq. (14).

Reviewer (1.4)

The largest system size considered is $N_e = 30$, which is reasonable for a proof-of-concept but leaves open questions regarding scalability and computational costs. A discussion of how computational costs scale with increasing system size would provide a clearer picture of the practical feasibility of this approach for larger quantum systems.

Response

We agree with the Reviewer that scalability is an important point. We hope that from the previous answers it is now clear that what we propose *does not* limit scalability, but in fact it may actually improve it. For completeness, we address scalability from three perspectives below: the general methodology, the limitations of the current physical model, and the scaling of the architecture and sampling algorithm.

1. **(General scalability from the perspective of the methodology):** What we

Figure 2: Convergence of the ground state energy per site for HF-TQS (a) and of α (b) as a function of epochs for $N_e = 60$ at $t/U = 0.16$. (c) Histogram indicating the total relative frequencies R^j , according to Eq. (15) of the main text, for the excitation classes $\mathcal{E}(s)$ from Eq. (14).

show is that for *any* system size N_e , when the dominant weight of the ground state of \hat{H} is in the low-energy part of the spectrum $E_0(\mathbf{s})$ of the effective theory \hat{H}_0 , this improves sampling efficiency and the ansatz expressivity (see *Fig. 1a*). This means that you can only gain computational efficiency, by construction, if the neural network only needs to consider (partially or entirely) the subspaces spanned by $C_k^{N_e} = \frac{N_e!}{k!(N_e-k)!}$ for a finite number of particle-hole excitation classes k .

2. (Scalability for the current physical model): The Hamiltonian we study is inspired by models where topological obstructions make it impossible to write symmetric local Hamiltonians in real space, requiring us to work in momentum space. This means that any momenta can be coupled—if they satisfy the conditions in Appendix A—introducing many more energetically relevant scatterings for certain t/U compared to local real-space models. This is why the advantage seen from using prior information about HF is quite non-trivial. To highlight this even more, in Fig. 2, we present additional results at $t/U = 0.16$ for $N_e = 60$ electrons considering the HF-TQS. From the histogram, one can see that the number of final unique states is bounded by $n_U^f < C_2^{N_e=60}$, where $C_k^{N_e} = \frac{N_e!}{k!(N_e-k)!}$. For the other effective theories, to achieve the same precision, one would have to set n_U to the order of 2^{60} . This is an impossible limit to reach due to memory constraints. A very naive estimate, without considering how the Transformer scales with memory, assuming all these unique states are being stored as float64 (8 bytes) variables would be in the order of million terabytes. This is a drastic improvement for scalability in this model, in our honest opinion.

3. (Computational costs for Sampling and Transformers): It should be clear that the Transformer here scales as in other approaches, e.g., D. Luo et al.: Phys. Rev. Res. 5, 013216 (2023). The batch autoregressive sampler scales with $\mathcal{O}(n_U N^3/3)$ (see text below *Fig. 3* in Y. Wu et al.: arXiv:2306.16705 (2023)).

Note that the scaling for the effective theory depends on its nature. For Hartree-Fock, it typically scales polynomially with system size (see e.g. J. Hermann et al.: arXiv:2208.12590 (2022)). The sampling procedure in Point 3 is definitely improvable. A very promising approach seems to be considering sampling without replace-

ment as in A. Malyshev et al.: arXiv:2408.07625 (2024).

To demonstrate that larger system sizes are achievable, we have included the results from Fig. 2 in our revised supplement and refer to it in the main text.

Reviewer (1.5)

The manuscript briefly describes the model architecture and optimization parameters, but additional details on how these were selected would improve clarity.

Response

We have explained in detail in Appendix C how the Transformer hyperparameters were chosen in distinct regimes of the phase diagram.

Reviewer (1.6)

Minor comments: Some text in the figures is too small to be legible. The boxes in Fig. 3(a)(c) should be more clearly explained, and the scale of the y-axis in Fig. 3(c) is somewhat unclear.

Response

We have changed Figs. 3(a) and (c) accordingly. In the new version, instead of using the boxes, we explicitly write in Fig. 3c the value of the coupling t/U in the three regimes: insulating, critical, and metallic. In addition, we have increased the text size in the subplots and boxes, which had smaller font size in comparison to the one in the main plots. We have also slightly changed the structure and presentation of all Figures, this should also help with the font-size.

Reviewer (1.7)

Minor comments: The code appears to be complete, but there is no instruction on how to run it. An updated README file with detailed instructions on how to reproduce the results in the main text would be helpful.

Response

We thank the Reviewer for the feedback. We have updated the “README” file accordingly.

Responses to Reports 2 and 4

Reviewers (2,4)

In “Physics-Informed Transformers for Electronic Quantum States” by Sobral et al., the authors introduce a new approach to electronic quantum states search via innovative usage of autoregressive variational wave-functions. The presented method offers a degree of interpretability which in particular presents a step forward. This work combines very well a reasonable level of physics insight, while still keeping the generic approach of neural network quantum states. The paper is well written, and results are understandably presented. Overall, we are positive about this manuscript and would recommend it for publication pending the comments listed below. (2.1)

Response

We appreciate the Reviewers’ understanding of our work’s main points and their recognition of its contributions to the field. We have implemented the suggested modifications and hope these revisions address their concerns comprehensively.

Reviewers (2,4)

Line 49: “TQS ... given their previously mentioned advantages” Can the authors clarify these advantages and over what they are? It is stated that the transformer quantum states “have proven effective in providing highly accurate representation of ground states...”, but it is not clear what the authors are really comparing the transformer to, here, and what evidence they have for these benefits. (2.2)

Response

We thank the Reviewers for pointing this out, it is indeed confusing. We wanted to refer to the paragraph starting in line 19, but “advantages” is not a good word here. Instead, we have just removed this sentence, since the previous one is sufficient.

Reviewers (2,4)

Line 71-78: This is just a difficult sentence to read, could the authors split it and simplify? (2.3)

Response

We have revised it accordingly. We thank the Reviewers for bringing this to our attention.

Reviewer (2.4)

Lines 182-189: Again, can this part be rewritten for clarity. Especially to better explain how the neural network is used to produce both outputs. This text alone is not entirely sufficient to make this point clear. The figure the text is referring to is very clear, so perhaps the solution is just to connect the two better.

Response

This is indeed a good suggestion. We have rewritten this paragraph in more detail in the new version of the manuscript.

Reviewer (2.5)

Figures 2 and 3 are very dense as is the explanation of all the captions. Currently it takes much time and effort to understand what is plotted. We suggest creating more verbose captions to aid the understanding of the presented results.

Response

We have revised it accordingly. In addition, to improve clarity, we separated *Fig. 3* into two new Figures: *Fig. 3* now focuses only on the comparison between distinct bases, and *Fig. 4* (new) contains information about the fermionic bilinear observable from eq. 16 (old *Fig. 3c*). We hope that these modifications make the understanding of these results more straightforward. Once again, we thank the Reviewers for bringing this to our attention.

Reviewers (2,4)

Line 283: The slight inconsistency of nomenclature is a little confusing, the use of HF-TQS is valid and clear, but when this is used to refer to Figure 2, where this phrase is not used, it loses some of the clarity and makes it harder to easily flick from text to figure to see the point being made. Nonetheless, the result shown in Figure 2 is impressive. (2.6)

Response

This is true, we missed this typo. In the new version of the manuscript, the label of *Fig. 2a* correctly displays HF-TQS now instead of only TQS.

Reviewers (2,4)

Lines 290-296: The authors claim that this HF-TQS ansatz applies to larger systems, but this is not true for $t/U < 0.1$. This is addressed later in the text, but it seems odd not to mention it as a caveat when claiming that the method scales well. (2.7)

Response

We have completely rewritten this paragraph in the new version. There is a subtlety in this region that we wanted to highlight for transparency reasons, as explained in Section IIc. One could still apply the HF-TQS in the gray region, but it would need a higher number of unique states n_U in this region, a common feature observed in all bases.

In Fig. 1 we show one example with larger n_U for $N_e = 14$ at $t/U \approx 0.10$ to demonstrate this point. Since $n_U > 2^{14} > n_U^{\text{previous}} = 4000$, more corrections are found and α deviates more from unity close to the phase transition. Compare this histogram with the one at $t/U = 0.09$ in *Fig. 3c*. The final number of states n_U^f kept by the Transformer is actually very close to 2^{14} . However, the hierarchy over particle-hole excitation classes $\mathcal{E}(\mathbf{s})$ is maintained, contrasting with its absence in other effective theories (see *Fig. 3c*), with double particle-hole excitations still dominating the corrections. By consistency, similar results should apply for $N_e \geq 14$ in the gray region of *Fig. 2a*. Away from the critical regime, we still observed substantial advantage for all system sizes that we considered since $n_U^f \ll 2^{N_e}$. We added a remark about this in the new *Fig. 2* caption and separated the discussion focusing on the insulating-metallic regimes from the critical one to avoid further confusion.

Reviewers (2,4)

Figure 3: We recommend a log scale for the inset of panel (a). It is clear that this inset shows that the band basis is much less performant, but it would be also interesting to see the relative performance of the other two bases shown, and this is not clear from the linear scale. (2.8)

Response

We thank the Reviewers for the suggestion. We have produced a new version of the inset of *Fig. 3a* using the log scale. It is indeed much clearer to see the relative

performance between all bases now.

Reviewers (2,4)

(J) Lines 361-363: Please can the authors refer back to the evidence in their plots that support this claim, I think this would also help the reader switch from text to figures to convince themselves of the result. (2.9)

Response

We have revised it accordingly on the new version of the manuscript.

Reviewer (2.10)

Figure 4: The misalignment of the boxes is a little undesirable, it would make for a nicer figure to have consistent subplot size and placement.

Response

We agree with the Reviewers. We have created a new version of the figure with consistent subplot sizes and placement. We have reorganized *Fig.3c* in a similar way.

Reviewers (2,4)

Equation 15 and the surrounding text: the interpretation of R^i is a little unclear from first glance, a reader new to this topic would benefit from some interpretation of this quantity to give them something to compare to in their minds. (2.11)

Response

We understand that the physical meaning of R^j might be a bit difficult to grasp upon first reading the manuscript. In short, we first define equivalence classes of states \mathbf{s} in Eq. (14), labeled by the number j of particle-hole excitations $\mathcal{E}(\mathbf{s})$ with respect to the given spectrum of \hat{H}_0 . Then R^j is just the relative weight of the ground state wave function in each such sector with $j \neq 0$ particle-hole excitations, normalized such that $\sum_{j \neq 0} R^j = 1$. Note that the reference state with $j = 0$ is excluded in the sum here, such that R^j is the relative weight of the *corrections* to the reference state. More formally, let us define $\hat{P}_j = \sum_{\mathbf{s}} \delta_{\mathcal{E}(\mathbf{s}),j} |\mathbf{s}\rangle \langle \mathbf{s}|$ to be the projector onto the subspace of states with $\mathcal{E}(\mathbf{s}) = j$; we then have $R^j = \|\hat{P}_j |\Psi\rangle\|^2 / (1 - \alpha^2)$, where $|\Psi\rangle$ is the ground state. Intuitively, in our two-band model, these quantities provide a quantitative measure of which particle-hole excitation sectors contribute most to the

corrections needed to reach the true ground state from \hat{H}_0 . We have modified our discussion around Eqs. (14) and (15) to make this more clear to the reader.

Reviewers (2,4)

Lines 388-389: Why do the authors claim that this is the curse of dimensionality? (2.12)

Response

Curse of dimensionality here would mean that away from the insulating regime, the band and chiral bases need 2^{N_e} samples to accurately represent the ground state. We have explained this in more detail on the text as: *The chiral and band bases are fundamentally limited by the curse of dimensionality: the ground state physics cannot be captured by just a few dominant basis states, as demonstrated by the significant contributions of all R^j away from the insulating regime. This broad distribution would consequently limit their applicability for larger system sizes N_e .* We hope that this modification improves clarity.

Reviewers (2,4)

The code in the appended repository appears readable and complete and the key functions are sufficiently explained, but it is left to the user to navigate usage of this repository. We urge the authors to explain the file structure and example usage in the readme. (2.13)

Response

We thank the Reviewers for the feedback. We have updated the README file accordingly.

Reviewers (2,4)

Overall, most of our comments are related to the presentation rather than the substance. The results are interesting and useful to the community. The work is creating a useful synergy between injection of some physics knowledge while still leveraging the flexibility of the physics-agnostic methods. After the comments above are addressed we will be pleased to recommend this manuscript for the publication. (2.14)

Response

We thank the Reviewers for understanding precisely the main message of the work and for summarizing it so well with one sentence: “The work is creating a useful synergy between injection of some physics knowledge while still leveraging the flexibility of the physics-agnostic methods”. We hope that our modifications of the main text were sufficient to address the previous points, and we thank them once again the Reviewers for their great suggestions.

Responses to Report 3

Reviewer (3.1)

This paper introduces a physics-informed Transformer-based variational Monte Carlo approach for representing electronic quantum states. By selecting a computational basis that includes a physically motivated reference state—such as Hartree-Fock, the method enables efficient and interpretable wavefunction representations. The authors applied the method to a 1D fermionic model, demonstrating accuracy over generic basis choices. In addition, the authors analyze the interpretability of the approach by looking into how the variational wavefunction is close to the reference state. I have gone through the writing and the method in details. The work is solid and introduces an interesting approach. However, there are several crucial questions to be addressed to meet the publication standard of Nature Communication. The main concerns are on the scalability and performance of this approach compared to other state-of-the-arts approach in the field, as well as its physics contribution.

Response

We thank the Reviewer for going into detail over our work, and for considering it a solid and interesting approach. We address the Reviewer’s main concerns on scalability and performance in the next questions.

Reviewer (3.2)

Can the author elaborate a bit more how the autoregressive model is able to maintain the normalization by excluding the RS basis, is there any constraint to be imposed in the architecture?

Response

As long as one does not include the reference state energy contribution to the loss function, we have noticed that the Transformer always converges while obeying the normalization condition. For concreteness, we are showing one example in Fig. 3.

Here, the deviation of the normalization from 1 fluctuates on the scale 10^{-5} in the transient regime, since the reference state can have a residual number of occurrences from the batch-autoregressive sampler structure. Nevertheless, since this state is never incorporated in the loss function by construction, it converges to unity as the Transformer learns that it does not contribute to the ground state energy (since this is already taking into account by the term E_{RS} in eq. (9) of the main text.). The energy expectation value is always divided by this quantity in accordance with eq. (5) of the main text.

An alternative approach would be to directly impose the normalization constraint at each sampling step, similarly to what is done in the gauge checking procedure described in the suggested reference D. Luo et al.: Phys. Rev. Res. 5, 013216 (2023). What we are doing is somehow equivalent to “discarding” the RS samples, according to A. Malyshev et al.: arXiv:2310.04166 (2023). In their language (see Sec. IIIc in this Ref.), after convergence in Fig. 3, the sum of the occurrence numbers of the final samples *will* add up to N_s .

Figure 3: Normalization as a function of epochs for $N_e = 14$ at $t/U \approx 0.10$.

Reviewer (3.3)

Most of the calculations of the paper are done with electron number of order 10, and up to 30. However, many of the study in this field of using transformer quantum state can already scale up to 100 particles. For example, Luo, Di, et al. “Gauge-invariant and anyonic-symmetric autoregressive neural network for quantum lattice models.” Physical Review Research 5.1 (2023): 013216.: it computes 160 unit cells (maybe the author should also include this in the citation since it is an early work

that develops transformer quantum state) Viteritti, Luciano Loris, Riccardo Rende, and Federico Becca. “Transformer variational wave functions for frustrated quantum spin systems.” *Physical Review Letters* 130.23 (2023): 236401.: it computes 100 sites. Can the author comment why the current study is limited in system size and what the complexity and the computational time of this approach are.

Response

We thank the referee for bringing this point, which we also consider important. While we were familiar with this work, we inadvertently omitted citing it in our original manuscript. We added the paper to the introduction of the manuscript and thank the Reviewer for bringing this to our attention.

For scalability we address it from three perspectives below: the general methodology, the limitations of the current physical model, and the scaling of the architecture and sampling algorithm.

1. **(General scalability from the perspective of the methodology):** What we show is that for *any* number of electrons N_e , when the dominant weight of the ground state of \hat{H} is in the low-energy part of the spectrum $E_0(\vec{s})$ of the effective theory \hat{H}_0 , this improves sampling efficiency and the ansatz expressivity (see *Fig.1a*). This means that you can only gain computational efficiency, by construction, if the neural network only needs to consider a finite (partially or entirely) the subspaces spanned by $C_k^{N_e} = \frac{N_e!}{k!(N_e-k)!}$ for a finite number of particle-hole excitation classes k .

2. **(Scalability for the current non-local model):** The references cited present results for local models in real space, which are fundamentally different from ours. In our case, the Hamiltonian is inspired by systems like twisted bilayer graphene where topological obstructions make it impossible to write symmetric local models in real space, requiring us to work in momentum space. This means that any momenta can be coupled—if they satisfy the conditions in Appendix A—introducing many more energetically relevant scatterings for certain t/U compared to local real-space models. This is why the advantage seen from using prior information about HF is quite non-trivial. To highlight this even more, in *Fig. 2*, we present additional results at $t/U = 0.16$ for $N_e = 60$ electrons considering the HF-TQS. From the histogram, one can see that the number of final unique states is bounded by $n_U^f < C_2^{N_e=60}$, where $C_k^{N_e} = \frac{N_e!}{k!(N_e-k)!}$. For the other effective theories or in the conventional approach of, e.g., using the Bloch basis, to achieve the same precision, one would have to set n_U to the order of 2^{60} . This is an impossible limit to reach due to memory constraints. A very naive estimate, without considering how the Transformer scales with memory, assuming all these unique states are being stored as float64 (8 bytes) variables would be in the order of million terabytes. This is a drastic improvement for scalability in this model, in our honest opinion.

3. **(Computational costs for Sampling and Transformers):** It should be clear that the Transformer here scales as in other approaches, e.g., D. Luo et al.: *Phys. Rev. Res.* 5, 013216 (2023). The batch autoregressive sampler scales with $\mathcal{O}(n_U N^3/3)$ (see text below *Fig. 3* in Y. Wu et al.: arXiv:2306.16705 (2023)).

Note that the scaling for the effective theory depends on its nature. For Hartree-Fock, it typically scales polynomially with system size (see e.g. J. Hermann et al.:

arXiv:2208.12590 (2022)). The sampling procedure in Point 3 is definitely improvable. A very promising approach seems to be considering sampling without replacement as in A. Malyshev et al.: arXiv:2408.07625 (2024).

Reviewer (3.4)

The author mentions that the approach is different than neural network backflow in line 125, can the author comment how this approach performs compared to neural network backflow, we were aware of it but forgot mentioning it. For example, whether this approach is better than neural network backflow in 2d hubbard model.

Response

They are different, but not mutually exclusive. Neural network backflow (NNB) (Ref. [25] of the main text) is a powerful ansatz which restricts the wavefunction to determinant-based forms. We impose no such restriction. Our approach could very well be implemented together with NNB. In fact, in the context of Ref. [34] of the main text, orbital rotations on top of NNB seem to improve the ground state energy accuracy achieved for certain molecule internal angles γ (see Fig. 2b there) compared to NNB alone, though whether the effective theories considered here (HF, strong and weak coupling limits) would benefit distinct regions in the phase diagram of the 2D Hubbard model, for example, remains to be determined. An interesting research direction for the future, for sure, and we added a comment on it in the discussion section.

Reviewer (3.5)

The simulations of this paper is only in 1 dimension. However, the challenges of the fermionic simulation is in 2d or higher dimension. Can the authors comment or demonstrate whether this approach works well for 2d fermionic model?

Response

The approach should work in higher dimensions since it is based solely on the variational principle. Note that it would still be subject to other limitations that apply to standard neural quantum states approaches—such as the representational power of a given neural network architecture. Importantly, mean field theories become more accurate with increasing dimensions, meaning our 1D results represent the most challenging case for HF as an effective theory in this model, which does not have a clear exact solution for intermediate t/U . Consequently, in 2D (and even more so in 3D) we would expect HF to provide even greater advantages, since it should be more

accurate. We plan to investigate higher-dimensional models in future work. As this is also an interesting point, we added a comment in the discussion section.

Reviewer (3.6)

The authors mention that this approach is interpretable and show the effects of different choices of basis. However, it is not clear what the new physical insights can be learned from the results, since those limiting cases are usually expected in the corresponding regimes. Can the authors elaborate how this approach can be used to discover physics that is not known before?

Response

That’s a good question. First, we hope that our answers to previous questions were sufficient to clarify that the approach here is still rooted in the variational principle in a *complete* many-body basis, again as opposed to techniques like NNB. Therefore, this method is capable of “*discovering physics that is not known before*” as it can, for a sufficiently expressive neural network, represent a state distinct from the ground state of \hat{H}_0 . In fact, we have demonstrated that by also deliberately using the “wrong” \hat{H}_0 (chiral basis for weak interaction and band basis for strong interaction). To make the interpretability aspect more explicit, this comes at distinct levels:

(Validation of existing effective theories): We would like to reinforce that this is not only about a basis choice, since there is also an associated approximation to the energy spectrum which comes from a physically motivated effective theory – which surprisingly affects the training dynamics and internal representation of the TQS. *As physicists, we would like to determine domains of validity for these effective theories to both improve our understanding of their limitations and ultimately of the current phenomena being studied.* Our approach provides both, since we understand when and by how much these theories fail (with α deviating from one), and if they fail, only corrections on top of them are obtained. These may also have some degree of interpretability (as defined in the last section of the text). Also, although the limiting cases are very clear to physicists, seeing the Transformer using this information to build a more compact (and possibly meaningful) representation of the ground state corrections across most of the phase diagram, is highly non-trivial, in our view. Testing this as a proof of concept was also one of the motivations of why we chose the Hamiltonian in Eq. (1).

(Current Status of interpretability in NQS): Neural quantum states have revolutionized the way we use the variational approach for solving the many-body problem, but we have lost a significant portion of physical interpretability. In the past, physicists would come up with Ansätze that may even be the exact solution for certain problems, for example, (i) Gutzwiller projected wavefunctions within the Parton constructions formalism for the Haldane-Shastry model, (ii) Laughlin wavefunction in certain scenarios for fractional quantum Hall systems, etc. Naturally, this is an extremely challenging task for certain models. Outside of the exactly solvable scenario, these may be a good starting point as effective theories, for example. We have

constructed and demonstrated a very general framework where these can be used to guide a NQS as general as possible (i.e., without restricting the Ansatz hypothesis space). We anticipate that the synergy created here can also provide meaningful insights for proposing new, and possibly more complete (effective) theories.

List of changes made:

- Implemented all changes suggested by the Reviewers in the main text. They are highlighted in red for the resubmitted version.
- Updated README as advised by Reviewers **1**, **2**, and **4**.
- Added a citation to D. Luo et al.: Phys. Rev. Res. 5, 013216 (2023) in the introduction as advised by Reviewer **4**, as well as other relevant references cited in this response.
- Fixed inconsistent notation for wavefunctions $\Psi \rightarrow \psi$ in the SI, Appendix C, whenever appropriate.
- Changed *Figs. 2-4* to improve readability (as recommended by Reviewers **1**, **2** and **4**) and easiness of understanding. We also divided *Fig. 3* into two new figures. In particular:
 1. (*Fig. 2*) Increased the font sizes, improved notation, and extended the caption for clarity.
 2. (*Fig. 3*) Increased the font size, and replaced the squares in *Fig. 3c* to the actual coupling values t/U to improve clarity, as suggested by Reviewer **1**. We have removed the old *Fig. 3d* here to focus this panel on section “**D. Other effective theories**”. This also helps on increasing the font size for *Fig. 3c*.
 3. (*Fig. 3*) Created a new version of the inset plot of *Fig. 3a* with log scale, as recommended by Reviewers **2** and **4**. We have also made minor modifications to *Fig. 3c* box alignments.
 4. (*Fig. 4*) New version of *Fig. 3d* focusing only on the momentum-resolved fermionic bilinear observable.
 5. (*Fig. 5*) New version of the image with consistent subplot sizes and placement, as recommended by Reviewers **2** and **4**. It should also be easier to refer to *Fig. 3* with the new labels for each effective theory.
 6. (*Fig. 10-11*) Added both Fig. 1 and Fig. 2 to the supplementary information (SI).
 7. Minor changes on SI plots for consistency with figures on the main text.
- Formatted the main text and SI according to the Nature Communications format.

2nd round of Responses to Reviewers

NCOMMS-25-02094B: “Physics-informed Transformers for Electronic Quantum States”

October 15, 2025

Responses to Report 1

Reviewer (1.1)

I appreciate the authors’ detailed rebuttal and the substantial revisions they have already made. A few important issues still need attention before the manuscript is, in my view, ready for publication in Nature Communications. 1. The revised draft argues that the critical region is “intrinsically hard” and that the sample cap n_U must scale exponentially for HF-TQS to converge. That conclusion will be convincing only if the authors can show that no established algorithm performs well there. I recommend the authors include additional benchmarks obtained with a different state-of-the-art method (for example, tensor networks). Even a modest-size comparison would clarify whether the growth of n_U is a limitation of HF-TQS or a generic obstacle.

Response

We thank the referee for their previous suggestions again and for recognizing our efforts. We have removed the word “intrinsically” by rephrasing this sentence in order to be very clear that we are referring to the transformer results only.

Although the reviewer suggested that tensor networks could give an idea of the growth of n_U at the critical point, as it being a generic obstacle, there are some subtleties. As the Reviewer is also probably familiar with, MPS are known to be excellent descriptions for gapped phases in 1D Hastings, Matthew B. “An area law for one-dimensional quantum systems.” *Journal of statistical mechanics: theory and experiment* 2007.08 (2007): P08024., so we would expect this method to be ideal in the insulating regime of the model here. However, at the critical point, the entanglement area law may be violated (as the gap closes), which could induce the computational complexity of these algorithms to also grow with system size. We would then have to probably employ more sophisticated approaches such as Chepiga, Natalia. “Probing universal critical scaling with scan density matrix renormalization group.” *Physical Review B* 110.14 (2024): 144401. From this argument, we can see that generically whether the growth of n_U is a generic obstacle would probably demand comparison between distinct approaches in tensor networks/other quantum Monte Carlo techniques, which although being a very interesting question, falls beyond our scope here. Apart from the apparent confusion caused by our word choice (which we changed), we do not see how tensor networks would give any answer on the dependence of NQS

on physically inspired bases, which was the main point of the work, since ED/exactly solvable limits already served as a benchmark to validate the results we see for smaller system sizes.

Reviewer (1.2)

2. An adaptive or data-driven criterion for choosing n_U would be beneficial.

Response

We have provided a data-driven criterion for choosing n_U after the discussion of the previous results in the SI (section 6). Note that the tree exact sampling method we use which was introduced by *Malyshev et al.* (Refs. 38–39 in the main text) is inherently adaptive: tree nodes corresponding to states that do not significantly contribute to the ground state are progressively pruned during sampling. The parameter n_U controls the maximum number of unique configurations explored before this adaptive pruning begins. In practice and up to hardware (memory) constraints, for larger system sizes, we set n_U based on the trend observed for n_U^f in well-converged cases for smaller system sizes (e.g. $N_e = 10$). More concretely, if we noticed that in the metallic regime the corrections are dominated by double particle-hole excitations in $N_e = 10$, we make sure to set $n_U > C(N_e, 2) = \frac{N_e!}{2!(N_e-2)!}$ for bigger system sizes $N_e > 10$.

Reviewer (1.3)

3. The present study focuses on a custom 1D, non-local momentum-space model inspired by moiré systems. Readers may wonder whether locality or dimensionality change the conclusions. I recommend including at least a brief test on a local lattice Hamiltonian (e.g. 1D Hubbard or J1-J2 chain) with established results. I would be happy to recommend publication once these points are satisfactorily addressed.

Response

We thank the Reviewer for this insightful suggestion. Locality is indeed an interesting avenue to explore. We expect the method to remain effective for local models, as they require fewer local energy estimators (or scatterings), which should accelerate the energy estimation in each epoch. While local lattice models would naturally allow us to do such investigations, as the Reviewer noted, we can also leverage our framework by modifying the Coulomb interaction—see Fig. 1—to a local version in momentum space given by $V(q) = \exp(-|q|/\lambda)/(1 + q^2)$. Here, the parameter λ controls the degree of locality. For instance, the case $\lambda = 2$ (denoted $V_{l-2}(q)$) ensures that the potential magnitudes of the first neighbors are comparable to those of the original Coulomb potential, while $\lambda = 0.5$ (denoted $V_{l-1}(q)$) defines an effective

Figure 1: Previous Coulomb interaction in comparison to localized modifications. Dots represent accessible scattering vectors q for $N_e = 10$. The potential $V_{l-1}(q)$ is effectively a first-neighbor-only interaction for this system size.

Figure 2: Simulations with the $V_{l-2}(q)$ potential. Ground state energy per electron as a function of epochs in the band, HF and chiral bases for $t/U = 0.009$ (insulating), $t/U = 0.03$ (critical) and $t/U = 0.1$ (metallic). The histograms below each plot indicate the total relative frequencies R^j , for the excitation classes $\mathcal{E}(s)$.

first-neighbor-only potential.

For comparison with ED, we focused on $N_e = 10$ electrons for the following simulations. For different values of λ , we noted that the critical point was pushed to smaller coupling constants t/U , and thus the insulating regime in the phase diagram is much narrower. Starting from the metallic side ($t/U = 0.1$) for the potential $V_{l-2}(q)$, we have the same behavior as before, i.e., the “best basis” (with respect to accuracy and smaller n_V^f) is still HF, with double particle-hole excitations dominating the corrections, as can be seen from the plots in Fig. 2. For $t/U = 0.03$ (critical), we again see the weights being distributed among all states in the Hilbert space for all bases. Finally, for $t/U = 0.009$, we observed that the band basis continues to perform quite poorly in the insulating regime, as can be seen from the inset of $E(\theta, \alpha)/N_e$ vs. epochs, since it converges to a higher energy estimate $E_{\text{band}} \rightarrow 0$. We have also observed that the HF basis needs a more frequent preconditioner update for the SOAP optimizer, which we have provided in the new version of the SI.

Finally, we have also noticed that the effect of constraining the potential to first neighbors only—see $V_{l-1}(q)$ in Fig. 1—just pushes the insulator to even smaller t/U coupling parameters. In Fig. 3, we show that the Transformer is still able to find corrections beyond HF for all three bases in this local version of the model for $t = 0.009$. The histograms here (not shown) follow the same distribution as those for $t/U = 0.1$ in Fig. 2, indicating that for this model we are still in the metallic regime.

Figure 3: Simulations with $V_{l-1}(q)$ potential. Ground state energy per electron as a function of epochs in the band, HF and chiral bases at $t/U = 0.009$.

Responses to Reports 2 and 4

Reviewers (2,4)

We thank the authors for the revision of the manuscript. We are very happy with the implemented changes. Re-reading the manuscript, we arrived at one more minor point: figure 5: The authors claim that the structure of this latent space w.r.t $E(s)$ is an interpretable benefit. At the same time they discard the structure as not being able to be interpreted for the band basis. However, the structure is arguably the most clear in this case. The authors do suggest that "While some clear structure also emerges for the band basis, we emphasize that the labels $E(s)$ do not directly translate to the energetics of the states: as discussed above, the RS is never close to the ground states in any regime, such that the number of excitations E above it also does not present clear energetic relevance either" however (a) this downplays the amount of structure in the plot and (b) this only begs the question of if this latent space structure is useful in the first place; if the structure remains despite $E(s)$ not being a relevant quantity, then how can we see this 'structure w.r.t $E(s)$ ' as being a robust interpretable quantity of interest. It is of course not surprising and quite reasonable that the latent space cannot be interpreted in all instances, but we would still recommend adding one sentence along these lines to emphasize that the interpretability of the observed structure is not always guaranteed as underscored by the example above. Provided this minor correction is addressed, we are happy to enthusiastically accept this manuscript for the publication in Nature Comm. (2.1)

Response

We thank the Reviewers once again for the great suggestions in the previous round and for noticing this additional point which indeed requires a better explanation. Following their recommendation we added the following sentence in this paragraph: "...Hence, interpretability of these structures is not automatically ensured, as the above example illustrates."

Responses to Report 3

Reviewer (3.1)

I first would like to thank the authors for the reply and the update of the paper, which is pretty helpful for improving the paper. Indeed, I shared the same concern with the first reviewer on the novelty and the scalability of the approach. While the reply from the authors provide further elaboration, it is subtle whether the contributions meet the standard of Nature Communication yet. It is definitely a very solid work, but my current recommendation is also more towards a topically special journal (still open to change my perspective). Specifically, here are a few comments and it will be crucial if the authors can address the concerns. 1. I want to first thank the authors for the explanation of the normalization, which is pretty helpful.

Response

We thank the Reviewer for recognizing our efforts to improve the paper and for their suggestions once again. We are also glad that the explanation of the normalization is clearer now.

Reviewer (3.2)

2. The explanation of scalability could be subtle for several reasons: (1) (General scalability from the perspective of the methodology): this argument seems to rely on the assumption that the Hamiltonian should be close to an effective theory with a finite excitation. However, this may be quite special scenario and most strongly-interacting systems may fail to satisfy such properties. (2) (Scalability for the current non-local model) The author mentions “using the Bloch basis, to achieve the same precision, one would have to set nU to the order of 2^{60} ”. While the total Hilbert space is 2^{60} , it is not clear whether all those basis will be equally important. Thus, even using other basis, the neural network may still be possible to have important sampling efficiently on such Hilbert space (since the neural network may learn the important latent subspace). Definitely the approach from the authors is helpful to reduce the samples in certain regime (especially when effective theory exists), which is a nice contribution. However, the current results are not conclusive whether such approach is really better than other previous approach in general over different regimes. In particular, it seems that the effectiveness of the current approach relies on the existence of the effective theory, so that the neural network just requires to learn the difference of the exact solution and the reference state. However, it is not clear how well such approach will work if the effective theory and a good reference state do not exist (which could be common). On the other hand, when an effective theory with a finite excitation exists, probably it implies that the problem has a good latent space and the conventional approach may also work well. The lack of systematic comparison of the current approach with other previous approaches make it hard to evaluate the true effectiveness of the current approach. The concerns are that the current approach may be limited

in the regimes where effective theories exist.

Response

We respectfully disagree that the explanation of scalability is subtle overall. Either it has advantages when an effective theory is close enough or it performs exactly as a “conventional-transformer” NQS-based approach would work if not (see band-basis results or chiral-basis for large t/U).

The Reviewer’s concerns can be reframed as fundamental questions about effective theory construction itself: “When do we expect physics-informed approaches to provide advantages? Can we always construct meaningful effective theories?” These are precisely the open questions that motivated our work, and our framework provides systematic tools to address them.

- **On the ubiquity of effective theories:** You can always construct mean-field descriptions, and most systems have exactly solvable limits that provide natural reference states. For concrete examples: in twisted van der Waals materials, there are compelling reasons to believe Hartree-Fock approaches provide sufficiently accurate starting points (see Soejima et al., PRB 102, 205111 (2020)). Beyond this, variations of Gutzwiller projected wavefunctions for quantum spin liquids remain the theoretical gold standard, built upon 20+ years of analytical+numerical developments (see recent comparison with ViT in Maity et al., arXiv:2501.00096 (2024)).
- **On neural networks learning “important latent subspaces”:** The Reviewer correctly notes that “*even using other basis, the neural network may still be possible to have important sampling efficiently on such Hilbert space (since the neural network may learn the important latent subspace).*” This is absolutely true and not in contradiction with our results with the chiral basis for the small t/U limit. Our work demonstrates that transformer performance in finding these latent subspaces is fundamentally intertwined with basis choice—a crucial insight rarely discussed in the literature (apart from Ref. 34).
- “*However, it is not clear how well such approach will work if the effective theory and a good reference state do not exist (which could be common).*” **Reply:** We have directly addressed this concern through our band-basis results and chiral-basis for larger t/U , which demonstrate that when the reference state is poor (i.e., when no good effective theory exists), our method reduces exactly to conventional transformer-based approaches (from Eq. 4 of the main text there are no differences to what is done in the literature with $\alpha \rightarrow 0$). The Reviewer’s concern seems to stem from a misunderstanding that our approach somehow fails when good effective theories are not available—this is incorrect. When the Hartree-Fock or strong-coupling reference states capture little physics as well, the transformer simply learns the corresponding corrections as it would in any standard NQS approach (with α converging to 0). We have added a more explicit comment on this in the revised manuscript.

Reviewer (3.3)

3. In general, I really appreciate the authors' efforts. The development of the current approach and the exploration on the interpretability are pretty valuable. The current study is more algorithmic oriented compared to physics oriented (since it does not really target at a particular concrete physics system and study the physics of such model), it becomes important to evaluate its algorithmic contribution and significance. As the first reviewer also points out, the main techniques like transformer have been proposed before . There are also other techniques like NNB and conventional transformer ((e.g. <https://arxiv.org/abs/2208.12590>) are used to study fermionic systems. It becomes helpful if the authors can have a systematic study on comparing different approaches with the new approach to show that the new approach really has advantages in general over different regimes with or without effective theories. I believe the current work is already very valuable towards neural network quantum state community and definitely worth publication in a topically oriented journal. I also really appreciate and like the update from the authors on improving the paper. While I am open to change the perspective with more evidences provided, it is hard to gauge at the moment whether the current work meets the high standard of Nature Communication on significant breakthrough and the interest of broad audience before the above concerns are addressed.

Response

We appreciate that the Reviewer recognizes some of the implications of our work and our efforts to improve the manuscript and analysis. We also thank them again for their suggestions. It seems that some misunderstanding is still present and we take the opportunity for clarification:

- “...The current study is more algorithmic oriented compared to physics oriented (since it does not really target at a particular concrete physics system and study the physics of such model)...” – **Reply:** We hoped that our answers for Reviewer’s Question 3.6 made this clear. The sentence in parentheses is not accurate since the model we considered is concrete and we demonstrate its metal-to-insulator phase transition physics with HF, ED, HF+TQS and exactly solvable limits+TQS. All this is extensively explained in the SI. We therefore disagree with this assessment and maintain our view that this work has very good potential for interpretability. Notably, the Reviewer simultaneously describes our interpretability exploration as “pretty valuable” while criticizing the lack of concrete physics grounding, yet meaningful interpretability work precisely requires concrete physics systems to validate the methods, as we provide.
- “...As the first reviewer also points out, the main techniques like transformer have been proposed before . There are also other techniques like NNB and conventional transformer ((e.g. <https://arxiv.org/abs/2208.12590>) are used to study fermionic systems...” – **Reply:** Again, what we do reduces back to a conventional Transformer-based NQS when using the band basis (or any other

not-physics-informed basis choice where $\alpha \rightarrow 0$). See the discussions above. We have added a sentence to make this more explicit on the text. We have also already answered the Reviewer Question 3.4 about differences with NNB previously, which seems to have been missed.

- “...It becomes helpful if the authors can have a systematic study on comparing different approaches with the new approach to show that the new approach really has advantages in general over different regimes with or without effective theories...” – **Reply:** This is precisely what we have done with the different bases. In regimes where the effective theories do not perform well (chiral and band, for example) would be equivalent to traditional approaches in the NQS community when $\alpha \rightarrow 0$, as we explained above.

The whole point of the work was the combination of physically-inspired Ansätze (mean-field or exact-solvable limits, as it has always been done in the strongly correlated systems community) with the already existing formalism of neural quantum states, which the Reviewer has agreed that presents advantages and does take a step further in the interpretability direction. In our view, we honestly think this can be extremely valuable to the sub-communities in condensed matter physics, quantum chemistry and beyond, since as the Reviewer probably knows, variational methods/NQS are widely applicable (see a recent example here for nuclear context Parnes, Elad, et al. “Nuclear responses with neural-network quantum states.” arXiv preprint arXiv:2504.20195 (2025)). We believe these contributions, combined with our systematic treatment of both the successful regimes (where “good” effective theories exist) and the fallback regime (where they do not), represent exactly the type of methodological advance that merits publication in Nature Communications since it provides a new framework for systematic effective theory validation.

List of changes made:

- Changed caption in a few plots on the SI and main text from “Energy per site” to **Energy per electron** to avoid confusion with real space lattice models.
- Since the model is in 1D we have also removed bold from vectors in eq. (12-13) and subsequent discussions on the main text. We kept the previous notation on the SI for generality.
- Indicated in the main text that when $\alpha \rightarrow 0$, by construction from eq. (4) the whole formalism is the same as what is typically done in VMC simulations - as suggested indirectly by Reviewer **3**.
- Implemented the changes suggested by the Reviewers **2** and **4** in the main text for the PCA results for the band-basis. They are highlighted in **red** for the re-submitted version.
- Added two new citations of previous work that were missed in the first/second versions of the manuscript (Ref. 29-30 on the new version).
- Provided a data-driven criterion for choosing n_U on the SI as recommended by Reviewer **1**.
- Added further simulations related to locality as suggested by Reviewer **1** to the SI.